# Viral infection of cells within the tumor microenvironment mediates antitumor immunotherapy via selective TBK1-IRF3 signaling

Michael C. Brown [1], Mubeen M. Mosaheb[2], Malte Mohme[3], Zachary P. McKay[1], Eda K. Holl[4], Jonathan P. Kastan[5], Yuanfan Yang [6], Georgia M. Beasley[4], E. Shelley Hwang[4], David M. Ashley[1], Darell D. Bigner[1], Smita K. Nair [4] & Matthias Gromeier [1,2]✉

Activating intra-tumor innate immunity might enhance tumor immune surveillance. Virotherapy is proposed to achieve tumor cell killing, while indirectly activating innate immunity. Here, we report that recombinant poliovirus therapy primarily mediates antitumor immunotherapy via direct infection of non-malignant tumor microenvironment (TME) cells, independent of malignant cell lysis. Relative to other innate immune agonists, virotherapy provokes selective, TBK1-IRF3 driven innate inflammation that is associated with sustained type-I/III interferon (IFN) release. Despite priming equivalent antitumor T cell quantities, MDA5-orchestrated TBK1-IRF3 signaling, but not NFκB-polarized TLR activation, culminates in polyfunctional and Th1-differentiated antitumor T cell phenotypes. Recombinant type-I IFN increases tumor-localized T cell function, but does not mediate durable antitumor immunotherapy without concomitant pattern recognition receptor (PRR) signaling. Thus, virus-induced MDA5-TBK1-IRF3 signaling in the TME provides PRR-contextualized IFN responses that elicit functional antitumor T cell immunity. TBK1-IRF3 innate signal transduction stimulates eventual function and differentiation of tumor-infiltrating T cells.

[1] Department of Neurosurgery, Duke University Medical School, Durham, NC, USA. [2] Department of Molecular Genetics & Microbiology, Duke University Medical School, Durham, NC, USA. [3] Department of Neurosurgery, University of Hamburg Medical Center, Hamburg, Germany. [4] Department of Surgery, Duke University Medical School, Durham, NC, USA. [5] University Program in Genetics & Genomics, Duke University, Durham, NC, USA. [6] Department of Pathology, Duke University Medical School, Durham, NC, USA. ✉email: grome001@mc.duke.edu

Targeting intra-tumor innate immunity may overcome layered tumor immune-subversion by provoking natural, multifaceted immune responses. Numerous, diverse innate immune-stimulating strategies have been developed for cancer immunotherapy[1–3]. Acute viral challenge of malignant tumors, e.g. via recombinant viruses, may achieve both cancer cell killing and innate immune activation, thereby arousing adaptive anti-tumor immunity. Yet, the question of how "oncolysis" vs. innate immunity contribute to virotherapy, and the extent to which they occur in heterogeneous and immune-dysfunctional tumors, remains unclear. Moreover, the molecular mechanisms by which virotherapy empowers immune surveillance, and whether it differs from that of synthetic pattern recognition receptor (PRR) agonists, is not defined. Answering these questions is central to the clinical advancement of innate-stimulating immunotherapy.

Phase-1 testing of intratumoral PVSRIPO, a highly attenuated rhino:poliovirus chimera[4], revealed durable radiographic responses and a 21% survival rate in recurrent glioblastoma (GBM) patients at 36 months post-therapy, relative to 4% survival of a criteria-matched historical control cohort[5]. Further clinical testing in GBM, breast cancer, and melanoma[6] is ongoing. Beyond direct viral lysis of malignant cells, PVSRIPO has a peculiar, non-lethal virus:host relationship with dendritic cells (DCs) that is associated with sustained type-I/III interferon (IFN) secretion; instigating intratumor innate inflammation; and priming antitumor CD8 T cell immunity[7–9]. We sought to define the roles of 'oncolysis' vs. innate immunity in mediating PVSRIPO antitumor immunotherapy, and decipher the innate inflammatory nature of virally induced antitumor immunity relative to known Th1-promoting innate agonists.

In this work, we discover that PVSRIPO mediates antitumor immunotherapy via direct infection of non-malignant TME constituents, particularly macrophages, without overt tumor cell lysis. Virally induced innate immunity is mediated by MDA5-dependent, sustained TBK1-IRF3 signaling, is associated with exaggerated type-I/III IFN secretion, and diverges consistently from TLR-initiated NFκB dominant responses. Selective TBK1-IRF3 signaling via MDA5 engages systemic, polyfunctional antitumor T cell responses.

## Results

During pathogen infection, locoregional innate inflammation engages downstream immune responses[10]. It follows that cancer-targeting virotherapy and PRR agonists must function within the heterogenous and immune-distorted TME to accomplish both oncolysis/cytotoxicity and antitumor immunotherapy. Thus, we devised an ex vivo tumor tissue slice assay, retaining the mixed composition of the TME, to analyze tumor-intrinsic inflammatory and lytic events after PVSRIPO infection/PRR agonist treatment (Fig. 1a).

**PVSRIPO infection of tumor tissue primarily induces type-I/III IFN.** Since the poliovirus receptor CD155 is necessary and sufficient for PVSRIPO infection to occur, we first measured CD155 distribution in relevant human glioma-associated cell types. Tumor-associated macrophages/MDSCs (hereafter "TAMs") were distinguished from microglia using two distinct gating strategies[11,12]. Surface expression of CD155 was detected on various tumor-associated cell types: glioma tumor cells (CD45$^{Neg}$, CD31$^{Neg}$) expressed less CD155 relative to endothelial and myeloid cell populations, including TAMs, microglia, and neutrophils (Fig. 1b; Supplementary Fig. 1). We treated ex vivo glioma tumor slice cultures with PVSRIPO alongside known canonical type-I IFN inducing PRR agonists: Poly(I:C) [TLR3 agonist[13]]; Lipopolysaccharide [LPS; TLR4 agonist[14]]; and 2′3′-cGAMP [cGAMP; STING

agonist[15]] to define the inflammatory impact of engaging diverse PRR signaling pathways. Treatment with PVSRIPO (48 h) minimally affected overall cell viability indicating lack of widespread oncolysis, contrasting sharply with overt lysis of established GBM cell lines in vitro by the same virus dose (Fig. 1c). Rather, robust induction of (type-I) IFNα/β, (type-III) IFNλ1 and CXCL10 was detected in the slice culture supernatant (Fig. 1d). In contrast, LPS and Poly(I:C) mainly caused TNF, IL-1β, IL-6, and IL-10 secretion; cGAMP treatment led to CXCL10 and, to a lesser extent, type-I/III IFN release (Fig. 1d). In line with ex vivo GBM slices, PVSRIPO-infection-induced CXCL10 and type-I/III IFN responses in breast cancer and melanoma tissue slices (Fig. 1e). Thus, the dominant effect in ex vivo tumor tissue infected with PVSRIPO is a type-I/III IFN inflammatory signature in the TME, that is distinct from that induced by synthetic PRR agonists.

**TAMs induce IFN responses after PVSRIPO infection of tumor tissue.** Macrophages are natural poliovirus targets in susceptible primates[7,16], TAMs shape tumor immune function[17], and macrophages are sentinel pathogen detectors[18]. Hence, TAMs may be key to PVSRIPO/PRR-initiated inflammation. We treated tissue slices with PVSRIPO/PRR agonists (48 h), generated single-cell suspensions, and analyzed macrophage expression of activation markers by flow cytometry. TAMs (defined by CD45+, CD11b+, CD14+; Supplementary Fig. 1[11]) were the only myeloid cell population reliably recovered from tissue slices (Supplementary Fig. 3a). All treatments enhanced PD-L1 expression on TAMs (Fig. 1f), with variable induction of CD40 and HLA-DR (Supplementary Fig. 3b); PD-L1 expression on endothelial cells and tumor cells was either absent or unchanged. Since PVSRIPO provoked type-I/III IFN responses in tumor tissue (Fig. 1d, e), we analyzed cell-intrinsic IFN responses by staining for the IFN-stimulated gene, IFIT1. TAMs induced significantly more IFIT1 expression than other tested cell types (Fig. 1g; Supplementary Fig. 3c).

To directly address the role of TAMs in PVSRIPO/PRR agonist-induced cytokine responses, we generated single-cell suspensions from GBM samples, depleted CD14+ cells (Supplementary Fig. 4a), and treated mock- and CD14+-depleted suspensions with PVSRIPO, LPS, Poly(I:C), or cGAMP. Mock-depleted samples released cytokines similar to ex vivo tissue slices, while CD14+ cell depletion ablated this response to PVSRIPO (except IL-6; Fig. 1h); it also blunted the response to LPS, Poly(I:C), and cGAMP (Supplementary Fig. 4b). Purified CD14+ cells induced cytokine responses equivalent to that of mock-depleted samples (Supplementary Fig. 4b), confirming the direct capacity of CD14+ cells to respond to PVSRIPO/PRR agonists. PVSRIPO infection of healthy donor PBMCs emphasized that the dominant source for cytokine release was CD14+ monocytes, and not T/B cells (Supplementary Fig. 4c, d). Collectively, our data indicate that TAMs/myeloid cells are the primary contributors to the initial tumor-intrinsic inflammatory response after PVSRIPO infection, and PRR agonism in general.

**PVSRIPO-infected macrophages and DCs mediate antitumor efficacy.** Since macrophages largely explained cytokine responses to PVSRIPO in primary tumor tissue, we asked whether PVSRIPO targeting of macrophages alone, in the absence of neoplastic cell infection, mediates antitumor efficacy in mice. We used mice transgenic for human CD155 (CD155-tg) and PVSRIPO adapted for growth in murine cells (mRIPO)[7], to permit PVSRIPO infection of murine myeloid cells. First, we confirmed that mRIPO-infected adherent peritoneal exudate cells (PEC, >90% macrophage) from CD155-tg mice induced type-I IFN-dominant cytokine patterns (relative to LPS) and activation markers similar

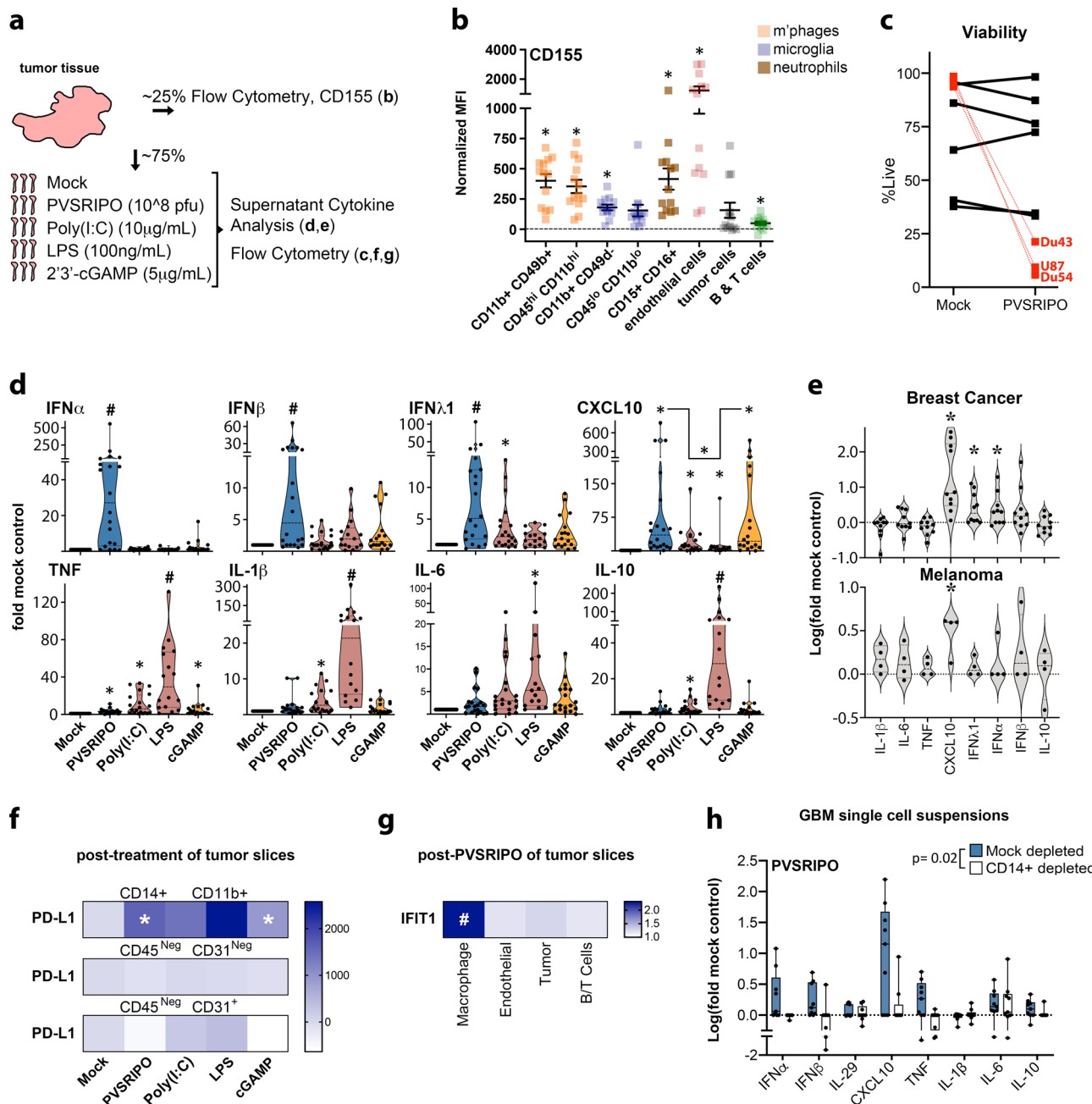

**Fig. 1 PVSRIPO treatment of ex vivo tumor tissue induces type-I/III IFN in TAMs. a** Fresh glioblastoma (GBM) tissue was analyzed for distribution of surface CD155 by cell type and responsiveness to innate stimuli. **b** GBM single-cell suspensions were analyzed for CD155 expression via flow cytometry ($n = 13$ tumors, see Supplementary Fig. 1 for gating); asterisks denote Bonferroni corrected one-sample $t$ test ($p < 0.0064$, two-tailed, from left to right: $p = 0.0001, 0.0001, 0.0001, 0.007, 0.0005, 0.0008, 0.03, 0.005$). Normalized MFI (median florescence intensity) was calculated by subtracting isotype control from stained MFI values for each cell type; mean ± SEM is shown. **c, d** Tumor slices were treated with PVSRIPO, Poly(I:C), lipopolysaccharide (LPS), or 2'3'-cGAMP as shown; pfu = plaque forming units. Post-treatment cell viability was measured after PVSRIPO treatment [7-AAD staining; $n = 6$ GBM, $n = 3$ glioma cell lines (DU54, DU43, and U87; mean of 3 experiments per cell line is shown)] (**c**) and fold-mock control cytokine release in supernatant was examined after all treatments (**d**; $n = 20$ for mock, PVSRIPO, and Poly(I:C); $n = 15$ for LPS; $n = 18$ for cGAMP). Only induced cytokines are shown; see Supplementary Fig. 2 for patient-specific induction; Tukey's post-hoc $p < 0.05$ (two-tailed) vs mock (*), unless otherwise indicated by bracket, or vs all other groups (#). **e** Ex vivo slice assay using breast cancer tissue ($n = 10$) and melanoma tissue ($n = 4$) testing only PVSRIPO as in (**d**); (*) one-sample $t$ test $p < 0.05$ (two-tailed; d: CXCL10 $p = 0.003$, IFN-λ1 $p = 0.02$, IFN-α $p = 0.03$; e: CXCL10 $p = 0.03$). **d, e** Violin plots present quartiles and median. **f, g** Dissociated, post-treatment tumor slices were analyzed by flow cytometry for PD-L1 expression [**f**, $n = 6$ GBM, normalized MFI—mock MFI is shown, (*) one-sample $t$ test $p < 0.05$ (two-tailed): CD14+ CD11b+ PVSRIPO $p = 0.02$, cGAMP $p = 0.02$] and intracellular IFIT1 expression for indicated cell types [**g**, $n = 4$ GBM and 2 breast cancer, fold-mock control MFI for each cell type is shown, (#) Tukey's post-hoc test vs all other groups: macrophage vs: endothelial $p = 0.0005$, tumor $p = 0.001$, B/T cells $p = 0.0004$]; see Supplementary Fig. 3 for gating and extended data. **h** Fresh ($n = 3$) and previously frozen ($n = 6$) GBM cell suspensions were subjected to mock or CD14+ depletion. Cell count-normalized mock- and CD14+-depleted suspensions were treated with PVSRIPO (48 h) and supernatant cytokines were measured; box shows median with quartiles and whiskers indicate range; $p$-value is from paired $t$-test (two-tailed) comparing the sum of log(fold-mock control) cytokine values; see Supplementary Fig. S4 for extended data.

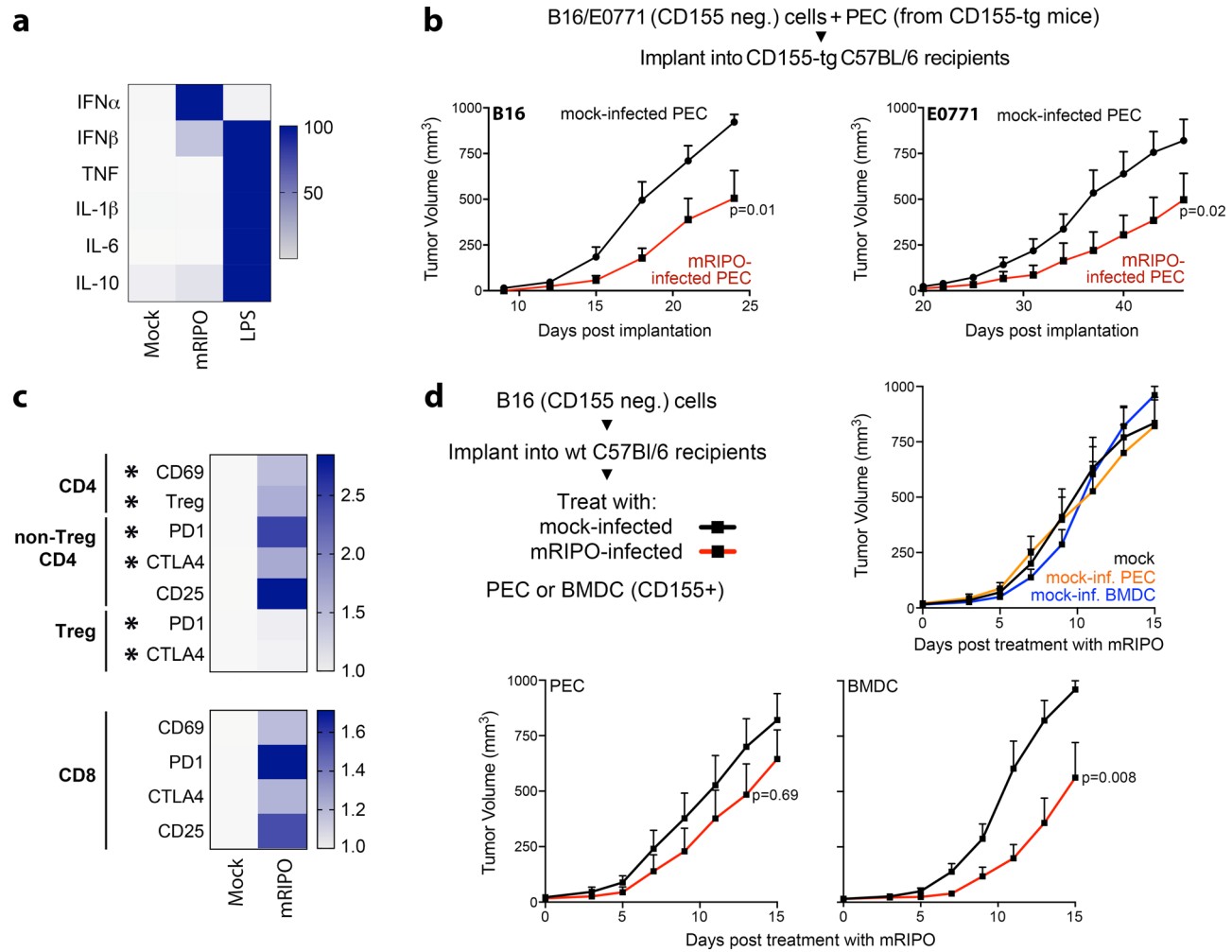

**Fig. 2 PVSRIPO-infected myeloid cells delay tumor growth. a** Peritoneal exudate cells (PEC) from CD155-tg mice were treated with mock, mRIPO [multiplicity of infection (MOI) of 10], or LPS (100 ng/ml) in vitro; cytokine release was measured and normalized to maximum value for each cytokine across all treatments (n = 2 experiments). **b** Wt B16 or E0771 cells were mixed with mock or mRIPO-infected CD155-tg PEC and implanted into CD155-tg mice subcutaneous or fat pad, respectively. B16: Mock n = 9, mRIPO n = 8; E0771: n = 9 for both. **c** E0771 tumor-bearing fat pads were harvested 21 days post E0771 + PEC (−/+mRIPO) implantation and analyzed by flow cytometry (n = 5/group); see Supplementary Fig. 5 for gating strategy. Asterisks indicate t test p < 0.05 (two-tailed; from top to bottom: p = 0.03, 0.01, 0.02, 0.02, 0.01, 0.02), values represent fold-mean mock %positive. **d** Wt mice bearing B16 tumors were injected with CD155-tg PEC or FLT3L-derived BMDCs after ex vivo pre-treatment with mock or mRIPO (MOI 10; 24 h) as shown; n = 9 mock and mRIPO-infected PEC, n = 8 PEC/BMDC (mock) and mRIPO-infected BMDCs. **b, d** Mean tumor volume + SEM is shown and p-values are from two-way ANOVA (two-tailed). All experiments were repeated at least twice and a representative series is shown.

to glioma tissue slices (Fig. 2a; Supplementary Fig. 5a). mRIPO-infected PECs exerted antitumor effects after co-implantation with wild type (wt, non-permissive to mRIPO) B16 melanoma or E0771 breast cancer cells in CD155-tg mice (Fig. 2b; Supplementary Fig. 5b). Infected PEC did not reduce tumor cell growth in co-culturing tests in vitro, indicating this was not due to direct tumoricidal effects (Supplementary Fig. 5c). Rather, antitumor effects were concurrent with T cell inflammation/ activation at the tumor site. E0771 tumor-bearing fat pad-infiltrating CD4 and CD8 T cells expressed elevated CD69, PD1, CTLA4, and CD25; with increased Tregs expressing inhibitory receptors PD1 and CTLA4 (Fig. 2c).

We next tested whether delivery of mRIPO-infected PEC or FLT3L-derived bone marrow-derived DCs (BMDCs) administered into established tumors mediated antitumor effects. PEC or BMDCs were derived from CD155-tg mice, treated with mock or mRIPO ex vivo (24 h), and then injected into established wt B16 tumors in wt mice (both lacking CD155). While administration of mock-treated PEC/BMDCs had minimal effects on tumor growth

(relative to PBS-injected tumors), transfer of mRIPO-infected PEC and BMDCs mediated tumor growth delay (Fig. 2d). Collectively, these data indicate that exclusive mRIPO infection of either macrophages or DCs in the TME delays tumor growth and mediates T cell inflammation.

**Infection of the TME mediates antitumor effects of PVSRIPO in mice.** Mouse CD155 does not permit poliovirus entry, but human CD155-tg mice are susceptible to infection and recapitulate paralytic poliomyelitis[19,20]. Expression of CD155 by (murine) B16 cells was sufficient to mediate cytotoxicity and viral protein expression in vitro (Fig. 3a, b). PEC from CD155-tg, but not wt (lacking human CD155) mice induced type-I IFN-dominant cytokine responses after mRIPO (Fig. 3c), associated with robust STAT1(Y701) phosphorylation (Fig. 3d). Consistent with the outcome of infection of primary human monocyte-derived macrophages (MDM)[7], viral translation was not detected, likely due to a robust host innate antiviral defense. UV-inactivated

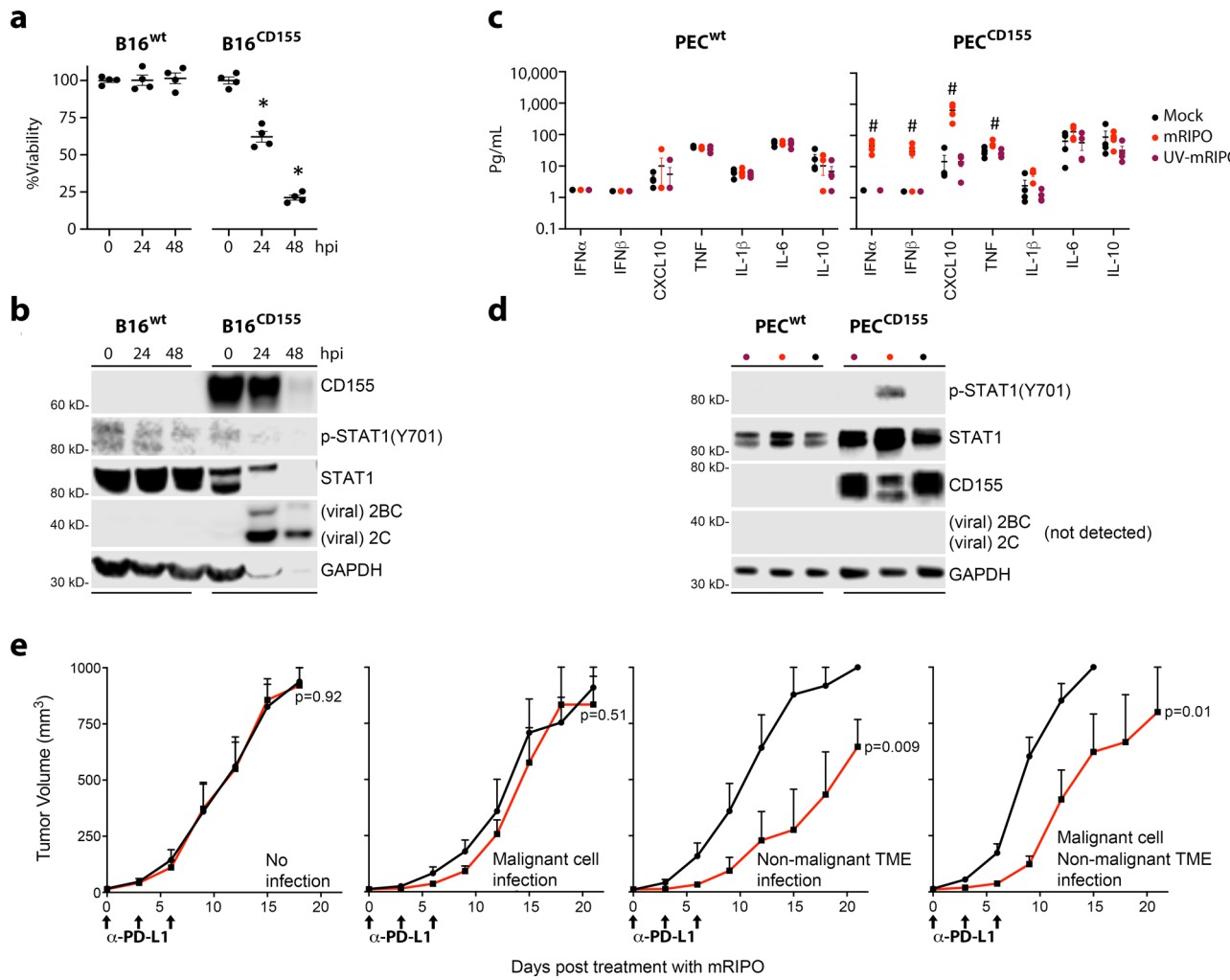

**Fig. 3 The non-malignant TME mediates antitumor efficacy of PVSRIPO in mice. a**, **b** B16^WT or B16^CD155 cells were treated in vitro with mRIPO (MOI 10). Viability (**a**) was measured by WST1 assay, values were normalized to %mean "0" time point, and immunoblot analysis (**b**) of cell lysates for CD155, STAT1, and viral protein (2C, 2BC). **c**, **d** PEC from WT (PEC^WT) or CD155-tg (PEC^CD155) mice were treated with mock, mRIPO (MOI 10), or UV-inactivated mRIPO (MOI 10 pre-inactivation titer) in vitro (48 h). Supernatant cytokines (**c**) were measured and immunoblot analysis (**d**) was performed as in (**b**). **a**–**d** n = 4 experiments, (**a**, **c**) mean −/+ SEM are shown, symbols (* and #) denote significant Tukey's post-hoc test (p < 0.05, two-tailed) compared to "0" time point (**a**, *, p < 0.0001 for both) or all other groups (**c**, #). **e** Wt or CD155-tg B16 cells were implanted into wt or CD155-tg mice to test (left to right): no infection (wt cells & hosts, n = 6 mock, 8 mRIPO), malignant cell only infection (B16-CD155-tg, wt hosts, n = 6 mock, 8 mRIPO), TME only infection (wt cells, CD155-tg hosts, n = 7 mock, 6 mRIPO), or TME + malignant cell infection (both CD155-tg, n = 9 mock, 8 mRIPO). Mice were treated intratumor with PBS or mRIPO (1 × 10^7 pfu) on day 0; PD-L1 blocking antibody was injected i.p. in all groups as shown. Mean tumor volume (mm3) + SEM are shown at each test interval; p-values are from two-way ANOVA (two-tailed); a representative series from two experiments is shown.

mRIPO failed to generate type-I IFN responses/STAT1(Y701) phosphorylation (Fig. 3c, d). We next tested antitumor efficacy of mRIPO (in the context of anti-PD-L1 antibody) after infection restricted to either malignant cells, the non-malignant TME, or both. This was accomplished by toggling tg-CD155 expression in tumor cells vs. the non-malignant TME (Fig. 3e). As expected, mRIPO therapy had no effect in models devoid of CD155, indicating mRIPO antitumor efficacy requires viral infection. Tumors in wt hosts implanted with CD155+ B16 cells yielded a transient antitumor effect (Fig. 3e). In contrast, treatment of CD155-tg hosts bearing wt B16 tumors caused durable antitumor efficacy; the outcome was similar with CD155+ tumors implanted in CD155-tg hosts (Fig. 3e). Likewise, injection of wt E0771 cells with mRIPO in the fat pad of wt mice did not delay tumor growth. However, the same procedure in CD155-tg mice resulted in marked tumor growth delay (Supplementary Fig. 5f). Thus, the antitumor efficacy of PVSRIPO is predominantly due to its capacity for infecting non-malignant TME constituents in mice.

Since assays in primary ex vivo human tumors indicated TAM-dependent innate inflammation in the absence of overt tumor cell lysis (Fig. 1), and myeloid cell infection alone was sufficient to mediate antitumor efficacy (Fig. 2), we conclude that PVSRIPO primarily exerts antitumor immunotherapy via innate immune activation in the TME. Therefore, we next sought to define the nature of PVSRIPO-induced innate inflammation in myeloid cells.

**Cytoplasmic dsRNA induces a distinct innate inflammatory pattern.** GBM[12] and breast cancer[21] TAMs prominently originate from peripheral monocytes. DCs, while rare in the TME, are key engagers of antitumor CD8 T cells[22,23], and are activated after infection with PVSRIPO[7]. Thus, we tested innate immune responses in both MDMs and DCs in vitro. We expanded our study to encompass an enterovirus related to PVSRIPO [Coxsackievirus A21 (CAV21)], PRR agonists [Poly(I:C); LPS; and cGAMP], the CSF1R inhibitor BLZ945[24], and IFNα2. PVSRIPO and CAV21

induced similar cytokine profiles, marked by type-I/III IFNs (Fig. 4a), and surface markers of activation (Supplementary Fig. 6c). In contrast, TLR agonists [Poly(I:C) and LPS] predominantly triggered TNF and IL-6 secretion; cGAMP, BLZ945, and IFNα2 minimally impacted MDM phenotype (Fig. 4a; Supplementary Fig. 6c). DCs responded similarly, with the exception that LPS modestly impacted activation phenotypes (Supplementary Fig. 6d–f). UV-inactivated PVSRIPO did not induce MDM activation (Supplementary Fig. 6g), indicating that PVSRIPO replication is required for innate activation. 'M2-like' macrophages[25] induced activation patterns similar to macrophages derived under standard conditions after PVSRIPO/PRR agonist treatment (Supplementary Fig. 6h). Collectively these findings indicate a consistently distinct, type-I/III IFN-dominant innate phenotype elicited by enterovirus replication.

Divergence of PVSRIPO from LPS activation patterns was consistent over a time course (Fig. 4b) and dose range (Fig. 4c; Supplementary Fig. 6i) in MDMs. Transfection (tfx) of Poly(I:C), to mimic cytoplasmic dsRNA, partially mirrored the pattern of PVSRIPO-mediated MDM activation, in particular, higher induction of IFNs despite less TNF and IL-6 relative to un-complexed Poly(I:C) (Fig. 4c; Supplementary Fig. 6i, k). Analogous to MDMs and tumor tissue slices, DCs treated with PVSRIPO or Poly(I:C)-tfx generated more type-I/III IFNs and CXCL10 relative to other stimuli; LPS only affected DC phenotype at a high dose (Fig. 4d; Supplementary Fig. 6j). As an indication of distinct innate activation patterns via intracellular dsRNA sensors [PVSRIPO/ Poly(I:C)-tfx] vs TLRs agonists [Poly(I:C)/LPS], t-SNE plots using surface activation markers yielded overlapping clusters of PVSRIPO and Poly(I:C)-tfx activated MDMs/DCs, that were separate from Poly(I:C)- and LPS-treated MDM/DC clusters (Fig. 4e). To test whether engaging cytoplasmic viral RNA innate receptors generically explains the inflammatory profile of PVSRIPO in GBM tissue slices (Fig. 1), we compared PVSRIPO treatment to that of CAV21 and Poly(I:C)-tfx. Indeed, engagers of cytoplasmic dsRNA sensors uniquely induced IFNα, while un-transfected Poly(I:C) and LPS provoked stronger release of TNF and IL-10 (Fig. 4f). T-SNE plots using cytokine data confirmed the clear separation between these agent classes in GBM tissue slices, MDMs and DCs (Fig. 4g). Thus, PVSRIPO, and intracellular dsRNA in general, induce a distinct type-I/III IFN-defined innate activation profile in myeloid cells.

**Cytoplasmic dsRNA induces sustained IRF3 activation and ISG expression.** To molecularly define divergent macrophage activation patterns, we analyzed innate signaling in MDMs, comparing PVSRIPO and LPS; IFNα2 treatment was added to delineate autocrine/ paracrine IFN-related events (Fig. 5a). PVSRIPO translation (viral 2C expression) peaked early (8 hpi) and faded thereafter, consistent with prior observations of viral translation restrained by host innate defenses in infected myeloid cells[7,8]. P-IRF3(S396)–a marker of IRF3 activation—was potently induced in PVSRIPO-infected MDMs, but less responsive to LPS (Fig. 5b). Enhanced p-IRF3(S396)-associated induction of ISG proteins (IFIT1, PKR, OAS1, ISG15), occurred mainly in PVSRIPO-infected MDMs relative to LPS-treated MDMs (Fig. 5b; Supplementary Fig. 8). PVSRIPO and LPS generated similar levels of p-p65(NFκB) (S536) and (NFκB inhibitor) IκBα decay. Yet, relative to PVSRIPO, LPS treatment caused substantially higher expression of NFκB proteins (eg. p105, p50, p100, p52) and the NFκB inhibitor A20 (Fig. 5b; Supplementary Fig. 8). These proteins are known targets of the NFκB transcriptional program[26]. Activation of IRF3 and NFκB were observed after PVSRIPO and LPS, but not IFNα2 treatment (Fig. 5a; Supplementary Fig. 8). Comparing the effects of plain vs. transfected Poly(I:C) with PVSRIPO revealed that

sustained IRF3 activation is due to cytoplasmic dsRNA, and does not occur with TLR3/4 agonists (Fig. 5c). A similar scenario unfolded in DCs (Fig. 5d). Thus, cytoplasmic dsRNA provokes robust IRF3 activation/ISG expression, relative to LPS/Poly(I:C), which preferentially induced NFκB signaling.

**MDA5-TBK1 signaling explains sustained IFN responses to PVSRIPO.** The innate antiviral response to picornaviruses is coordinated by MDA5, a PRR recognizing cytoplasmic dsRNA[27,28]. To test if MDA5 mediates the PVSRIPO activation profile, we compared PVSRIPO to Poly(I:C) and LPS in MDMs with or without MDA5 depletion (Fig. 6a, b). Immunoblots confirmed incomplete MDA5 depletion in our assay [MDA5 is an ISG and, thus, induced by PVSRIPO infection[29]] (Fig. 6a). Partial MDA5 depletion neither enhanced viral translation in MDMs nor induced viral cytopathogenicity (Supplementary Fig. 9a); it reduced p-STAT1(Y701) levels and the induction of ISG15 (Fig. 6a). Despite only modest depletion, MDA5 knockdown significantly tempered the PVSRIPO cytokine signature (IFNα/β, IFNλ1, CXCL10), but did not substantially alter Poly(I:C)/LPS-induced profiles (Fig. 6b).

PRRs elicit pro-inflammatory gene expression via two innate signaling nodes that drive elaborate transcriptional networks: TBK1, which along with IKKε phosphorylates IRF3, and the IKKα/β kinases that activate NFκB signaling[30,31]. PVSRIPO consistently induced p-IRF3(S396)-dominant patterns, with modest effects on NFκB subunits (Fig. 5). Thus, preferential activation of TBK1:IKKε may explain the type-I/III IFN-dominant profile of PVSRIPO-infected MDMs. To test this empirically, we compared untreated MDMs to MDMs treated with PVSRIPO or LPS in the presence of inhibitors of TBK1:IKKε [Bx795[32]] or IKKα/β (IKK16). Bx795 abolished the induction of type-I/III IFNs, CXCL10, and TNF due to PVSRIPO infection (Fig. 6c); induction of IL-1β, IL-6, and IL-10 was not affected. Meanwhile, IKK16 only modestly impacted the PVSRIPO cytokine profile (Fig. 6c). Bx795 also dampened PVSRIPO induction of surface markers of activation, while IKK16 only modestly restricted induction of CD40, CD83, and PD-L1 (Supplementary Fig. 9b). Similar trends were observed after LPS treatment, where CXCL10, IFNλ1, and TNF, but not IL-1β, IL-6, and IL-10 were blocked in the presence of Bx795 (detectable type-I IFN was not induced by LPS). IKK16 treatment broadly reduced all cytokine responses to LPS (Fig. 6c). Treatment with Bx795, but not IKK16, enhanced PVSRIPO translation, blocked IFIT1 expression and reduced viability of infected MDMs (Supplementary Fig. 9c, d). This implies a role for TBK1 signaling in inducing ISGs and restricting viral translation and replication. Thus, the MDA5-driven, type-I/III IFN signature that defines PVSRIPO-mediated MDM activation is attributable to TBK1 signaling. In contrast, IKKα/β signaling largely drives activation of MDMs after LPS treatment. Collectively, our findings reveal differential TBK1 vs IKKα/β kinase activation patterns that underly divergent innate phenotypes elicited by PVSRIPO vs. TLR agonists.

**Virotherapy of the TME results in distinct T cell phenotypes.** We next tested whether distinct innate signaling patterns, i.e. TBK1-IRF3 (PVSRIPO) vs. IKKα/β (TLRs), ultimately culminate in altered antitumor responses. These studies were performed in CD155-tg mice implanted with wt B16 cells (lacking CD155, non-permissive to PVSRIPO infection/lysis), in order to define the role of mRIPO as a TME PRR-engager. Each agent produced antitumor efficacy in the context of PD-L1 blockade (Fig. 7a; Supplementary Fig. 10a).

To address differences in antitumor immunity following mRIPO vs. TLR agonists, we performed a time-course assay of

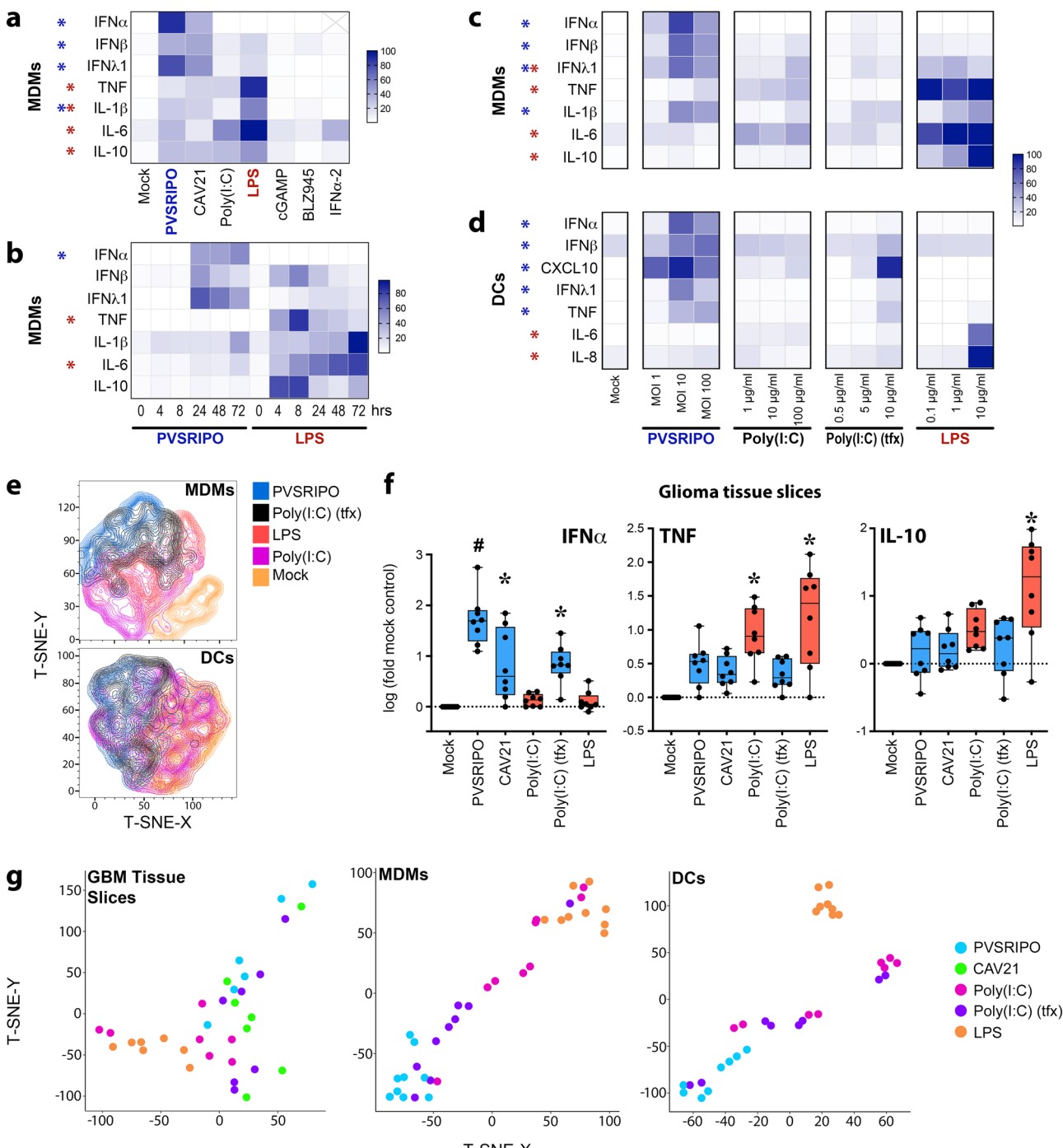

**Fig. 4 Cytoplasmic dsRNA induces a distinct innate immune activation profile. a** Monocyte-derived macrophages (MDMs) were treated with putative macrophage activators [PVSRIPO/CAV21, MOI 10; Poly(I:C)/LPS/cGAMP (as in Fig. 1a); BLZ945, 50 nM; IFNα2, 200 IU/ml] for 48 h; cytokine release was measured. Maximum cytokine concentration (pg/ml) was set to 100% for each cytokine; $n = 7$ experiments from three donors; IFNα values omitted for IFNα2 treatment. **b** Cytokine release after MDM treatment time course with PVSRIPO or LPS normalized as in (**a**); $n = 5$ experiments from 3 donors. (*) significant paired t-test comparing sums of PVSRIPO (blue) vs LPS (red) normalized (%max) values for all time points for each cytokine ($p < 0.05$, two-tailed; IFNα $p = 0.009$, TNF $p = 0.005$, IL-6 $p = 0.0007$). **c, d** Supernatant cytokines were measured 48 h after treatment of MDMs (**c**) or monocyte-derived DCs (**d**) with escalating doses of stimuli as shown; $n = 4$ experiments from 2 donors; normalized as in (**a**). **a, c, d** (*) indicates Dunnett's post-hoc test $p < 0.05$, two tailed, vs mock (**a**). **c, d** asterisks are shown if $p < 0.05$ at any dose. **a–d** Only treatment-induced cytokines are shown; asterisk color indicates significance for LPS (red) or PVSRIPO (blue); see Supplementary Figs. 6 and 7 for extended analyses and gating. **e** T-SNE plot based on surface activation marker staining after indicated treatments as shown for MDMs (top panel) or DCs (bottom panel). **f** Ex vivo treatment (48 h) of GBM slices was performed for denoted stimulants [PVSRIPO/CAV21 1×10^8 pfu; Poly(I:C)-tfx 2.5 μg/ml, Poly(I:C) 10 μg/ml, LPS 100 ng/ml], released cytokine concentration (pg/ml) was analyzed and normalized to log-fold-mock control for each cytokine ($n = 8$); box represents quartiles + median and whiskers indicate range; Tukey's post-hoc test $p < 0.05$ (two-tailed) vs mock treatment (*) or all groups (#). **g** T-SNE plots using fold-mock cytokine values after each treatment for GBM tissue slice culture assay (**f**), MDMs (**c**, middle doses), or DCs (**d**, middle doses).

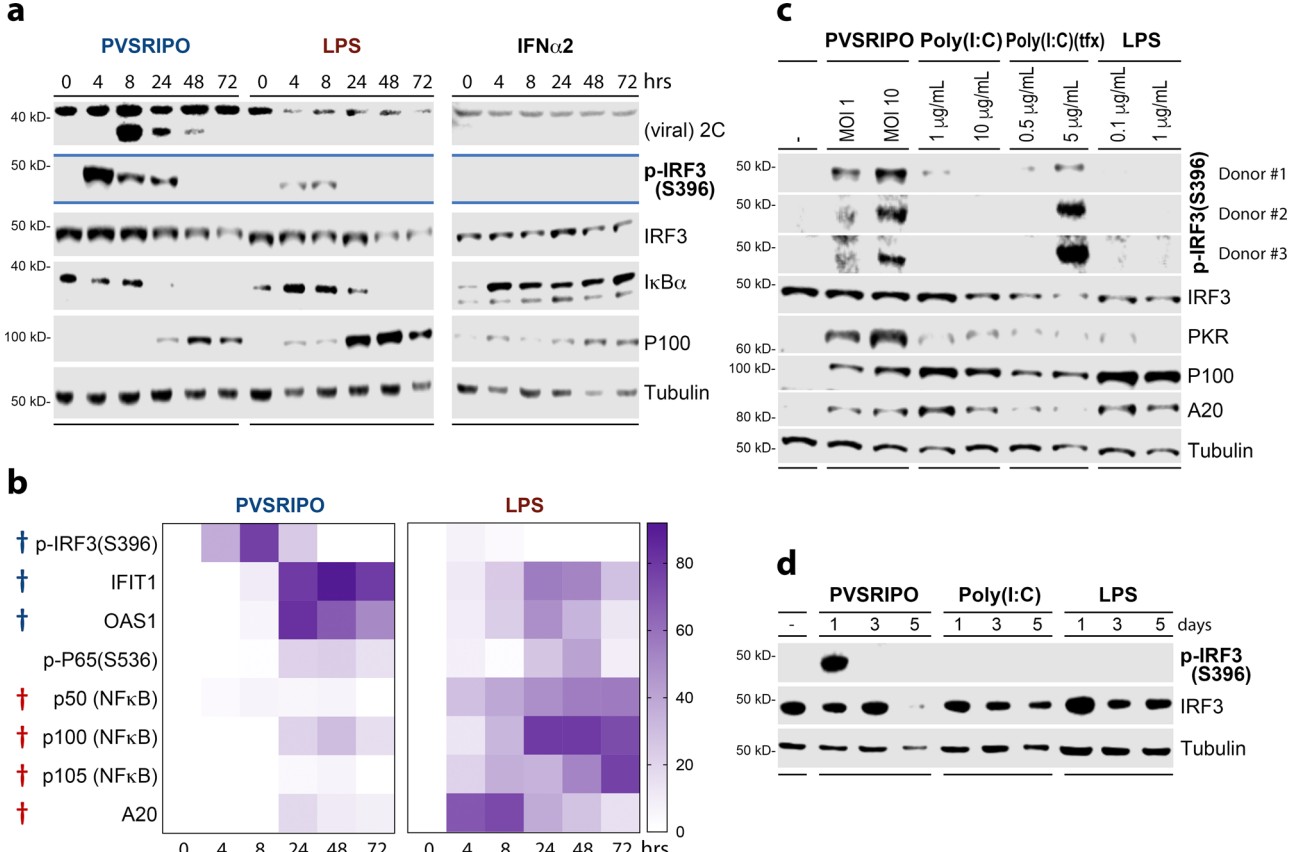

**Fig. 5 Cytoplasmic dsRNA causes robust, sustained p-IRF3(S396) and ISG expression. a, b** Immunoblot analysis of innate signaling in MDMs after treatment with PVSRIPO (MOI 10), LPS (100 ng/ml), or IFNα2 (200 IU/ml) over time. Additional analyses from four donors, including those quantitated in (**b**), are presented in Supplementary Fig. 8. **b** %Max densitometry values for each protein/phospho-protein followed by subtraction of zero time point is shown. (†) Significant paired *t* test (two-tailed) comparing maximum value (across time course) induced by PVSRIPO (blue) vs LPS (red); IRF3-p (S396) $p = 0.0001$, IFIT1 $p = 0.01$, OAS1 $p = 0.009$, p50 $p = 0.008$, $p100$ $p = 0.001$, p105 $p = 0.01$, A20 $p = 0.006$. **c** Comparison of PVSRIPO, Poly(I:C), Poly(I:C)-tfx, and LPS treatment using respective low and middle doses presented in Fig. 4c by immunoblot for relevant proteins. Three donors were tested, p-IRF3(S396) blots are shown for all donors. **d** DCs were treated with mock, PVSRIPO, Poly(I:C) or LPS along a time course for immunoblot analysis as shown; $n = 3$ experiments.

tumor-associated cytokine profiles and the TME after intratumor administration of each agent (Fig. 7b). Confirming TBK1 vs. IKKα/β dichotomy of mRIPO vs. LPS treatment, tumor homogenate cytokine analysis revealed stronger IKKα/β-driven cytokine induction by LPS (IL-6, TNF, IL-1β, IL-10), with CXCL10 (TBK1-dependent; Fig. 6) being the most prominently induced cytokine after mRIPO treatment (Fig. 7c). Intratumor Poly(I:C) induced a broader spectrum of inflammatory cytokines tested, including both TBK1 and IKKα/β-responsive cytokines (Fig. 7c). Notably, mRIPO treatment of murine tumors exhibited attenuated cytokine induction compared to TLR agonists, contrary to what was observed in human tumor slice cultures (Fig. 1). We suspect that this is a host range phenomenon, as the *enterovirus* genus of *picornaviridae* is a consummate human-specific taxon.

To compare how the tumor immune landscape reacts to mRIPO vs. Poly(I:C)/LPS-directed innate activation, tumors from mice treated as in Fig. 7b were dissected 1- and 7-days post-treatment for flow cytometry analyses. In line with stronger overall cytokine induction, LPS-induced significantly more neutrophil influx than mRIPO (day 1) (Fig. 7d). At day 7, influx of T and NK cells occurred after all treatments (Fig. 7d) coincident with enhanced expression of activation markers on macrophages [CD40, MHC II (IA/IE)] (Fig. 7e). Tumor-infiltrating CD4+ T cells induced by

mRIPO therapy expressed higher levels of the cytolytic effector granzyme B (GzmB) and T-bet [the Th1-promoting transcription factor that also supports T cell function[33]], and CD8+ T cells expressed higher levels of GzmB and IFNγ 7 days post-treatment (Fig. 7e; Supplementary Fig. 10). Other markers of activation/exhaustion were also elevated on tumor-infiltrating T cells after mRIPO (CD69, CTLA4, PD1, TIM3; Fig. 7e). Thus, despite—relatively—modest inflammatory cytokine release and subdued immune cell influx, mRIPO therapy of the TME proficiently mediated infiltration of GzmB/IFNγ+, Th1-polarized (T-bet-expressing) T cells.

**Virotherapy selectively induces functional antitumor T cell immunity.** Enhanced infiltration of CD8/CD4 T cells with functional phenotypes in tumors after mRIPO may be due to T cell responses against immunogenic poliovirus antigens, rather than tumor-specific features. To gauge whether tumor-infiltrating T cells (TILs) target tumor-specific antigens, we treated mice bearing B16.F10.9[CD155] tumors expressing OVA with PBS, mRIPO, or LPS. We did not include Poly(I:C), as it had inter-mediate effects on both TBK1 and IKKα/β signaling and T cell expression of GzmB/IFNγ (Fig. 7c, e). We implanted CD155-tg B16 tumor cells, for maximal viral translation to occur, to test whether abundant viral protein detracts from tumor antigen-

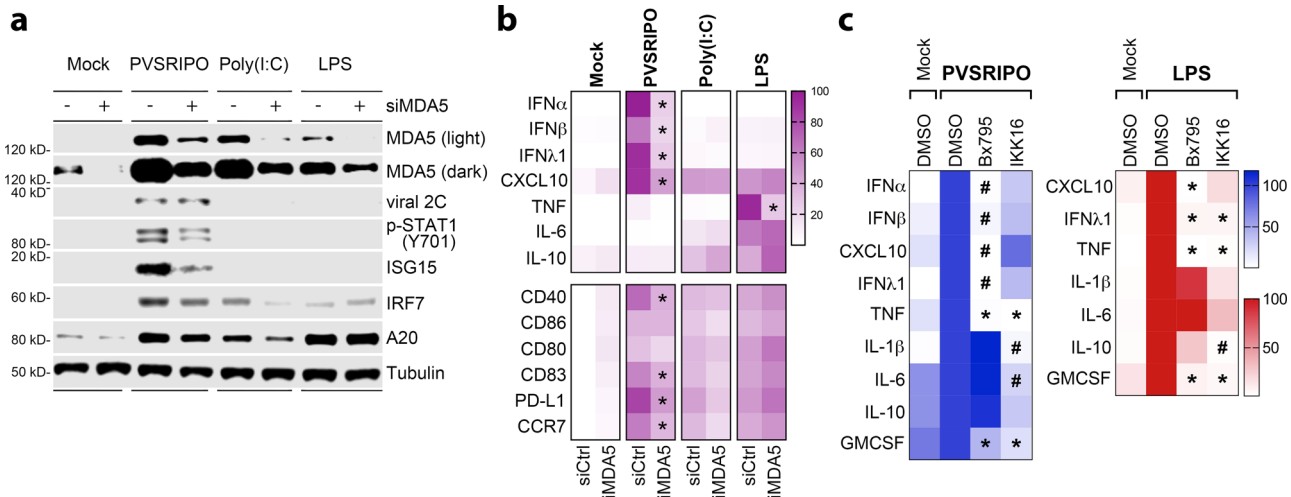

**Fig. 6 MDA5-TBK1 signaling explains the innate inflammatory response to PVSRIPO. a, b** MDMs were transfected with control siRNA or siRNA targeting MDA5 (48 h), followed by treatment with PVSRIPO, Poly(I:C), or LPS (48 h). **a** Immunoblot analysis of MDA5 depletion, viral protein 2 C, and other relevant proteins. **b** Cytokine release (top) and cell surface markers of activation (bottom) were measured. Fold-mock control siRNA cytokine concentrations were measured, geometric mean of %max induction values are presented; surface markers were normalized similarly, except baseline (mock control siRNA) %max values were subtracted and mean is shown. $N = 6$ experiments from two donors, asterisks indicate two-tailed Tukey's post-hoc test $p < 0.05$ comparing siMDA5 to siCtrl values for each treatment (PVSRIPO: IFNα $p < 0.0001$, IFNβ $p = 0.0001$, IFNλ1 $p < 0.0001$, CXCL10 $p = 0.0005$, CD40 $p = 0.0005$, CD83 $p = 0.006$, PD-L1 $p = 0.01$, CCR7 $p = 0.004$; LPS TNF $p < 0.0001$). **c** MDMs were pre-treated (1 h) with DMSO, Bx795 (2 µM; TBK1:IKKε inhibitor), or IKK16 (400 nM; IKKα:β inhibitor) followed by treatment with PVSRIPO (MOI 10) or LPS (100 ng/ml). Supernatant cytokines were analyzed; IFNα/β were not induced by LPS and thus were not included in the heat map; values were normalized from four experiments (two donors) by setting DMSO treatment values to 100% for each cytokine; geometric mean is shown; asterisks denote Tukey's post-hoc $p < 0.05$ (two-tailed) test vs DMSO control (*) or all groups (#) for each cytokine. See Supplementary Fig. 9 for extended data.

specific responses. Unexpectedly, post-mRIPO ELISpots revealed the presence of splenocytes secreting IFNγ without peptide stimulation (Fig. 8a, b). Subtraction of baseline IFNγ spot counts (Fig. 8b; left panel) revealed that both mRIPO and LPS treatment generated tumor antigen-specific (SIINFEKL) IFNγ+ T cells (Fig. 8b; middle panel). However, a larger average spot area indicated that splenocytes from mRIPO-treated mice secreted significantly more IFNγ on a per-cell basis (Fig. 8b; right panel); this difference was obvious on ELISpot plates, and was antigen-agnostic—i.e. differential IFNγ secretion was also evident at baseline and in concanavalin A positive controls (Supplementary Fig. 11). Together with the relatively enhanced GzmB/IFNγ+ expression in TILs after mRIPO treatment (Fig. 7e), we conclude that mRIPO therapy induces elevated splenocyte function compared to LPS.

Supernatant cytokines from splenocytes cultured alone (baseline; Fig. 8c), or co-cultured with B16.F10/B16.F10.9.OVA cells (Fig. 8c), were analyzed in parallel with ELISpots. At baseline, splenocytes from mRIPO-treated mice exhibited broad, significantly enhanced cytokine secretion (Fig. 8c). Co-culture of splenocytes with B16F10.9-OVA/B16.F10 cells revealed tumor cell-specific induction of a subset of Th cytokines (after subtraction of baseline cytokine secretion), with splenocytes from mRIPO-treated mice inducing higher cytokine secretion in each case (Fig. 8c). Splenocytes from mRIPO-treated animals provoked varied tumor-specific lymphocyte states as evidenced by expanded baseline and antigen-specific Th1- (IFNγ, TNF), Th2- (IL-5), and Th9- (IL-9) associated cytokine secretion. Thus, while mRIPO and LPS both induce antitumor immunity, they result in different functional states of antitumor T cells.

To directly compare antitumor function post-mRIPO vs. -TLR agonist therapy, mice bearing wt B16 tumors were treated with mRIPO, Poly(I:C), or LPS and tumor-draining lymph nodes (TDLNs) were harvested 7 days post-treatment. Relative to TLR agonists, TDLN cells from mRIPO-treated mice significantly

delayed B16 cell proliferation in vitro, as measured by carboxy-fluorescein succinimidyl ester (CFSE) dilution, and caused PD-L1 induction on B16 cells (Fig. 8d). These effects did not occur upon co-culture of TDLN cells from mRIPO-treated B16 tumor-bearing mice with heterologous CT2A mouse glioma cells, suggesting antigen-specificity (Fig. 8d). Analogous to enhanced baseline cytokine secretion observed in splenocytes (Fig. 8c), TDLN cells from mRIPO-treated mice exhibited higher IFNγ and IL-2 secretion alone and in co-culture with B16 cells (Supplementary Fig. 12c), possibly explaining the induction of PD-L1 on B16 cells after co-culture. To confirm the differential in antitumor T cell function after virotherapy vs. TLR agonists in vivo, we adoptively transferred splenic T cells from treated tumor-bearing mice to naïve recipients implanted with B16 tumors. Transfer of T cells isolated from mRIPO-treated mice had pronounced antitumor effects (Fig. 8e, f). Thus, leveraging selective TBK-IRF3 innate activation via mRIPO elicits a more functional antitumor T cell response.

**TBK1-IRF3-IFN signaling stimulates antitumor efficacy and TIL phenotype.** We next tested if TBK1-IRF3-driven innate signaling induced by mRIPO therapy explains polyfunctional T cell phenotypes by co-delivering mRIPO with mock (DMSO), the TBK1:IKKε inhibitor Amlexanox (AMX), or the IKKα/β inhibitor IKK16 (Fig. 9a). AMX blocks TBK1-IRF3 innate phenotypes in FLT3L-derived BMDCs (Supplementary Fig. 13a). Combined with mRIPO therapy, AMX, but not IKK16, attenuated intratumor T cell expression of GzmB and T-bet, particularly in CD8 T cells (Fig. 9a) and broadly reduced CD8 T cell, Treg, and NK cell infiltration (Supplementary Fig. 13b). The antitumor efficacy of mRIPO was also abolished by TBK1 inhibition (Fig. 9b). Thus, TBK1:IKKε innate signaling controls infiltration of GzmB and T-bet-expressing T cells and mediates antitumor efficacy after mRIPO treatment. Given the role of TBK1-IRF3 in inducing

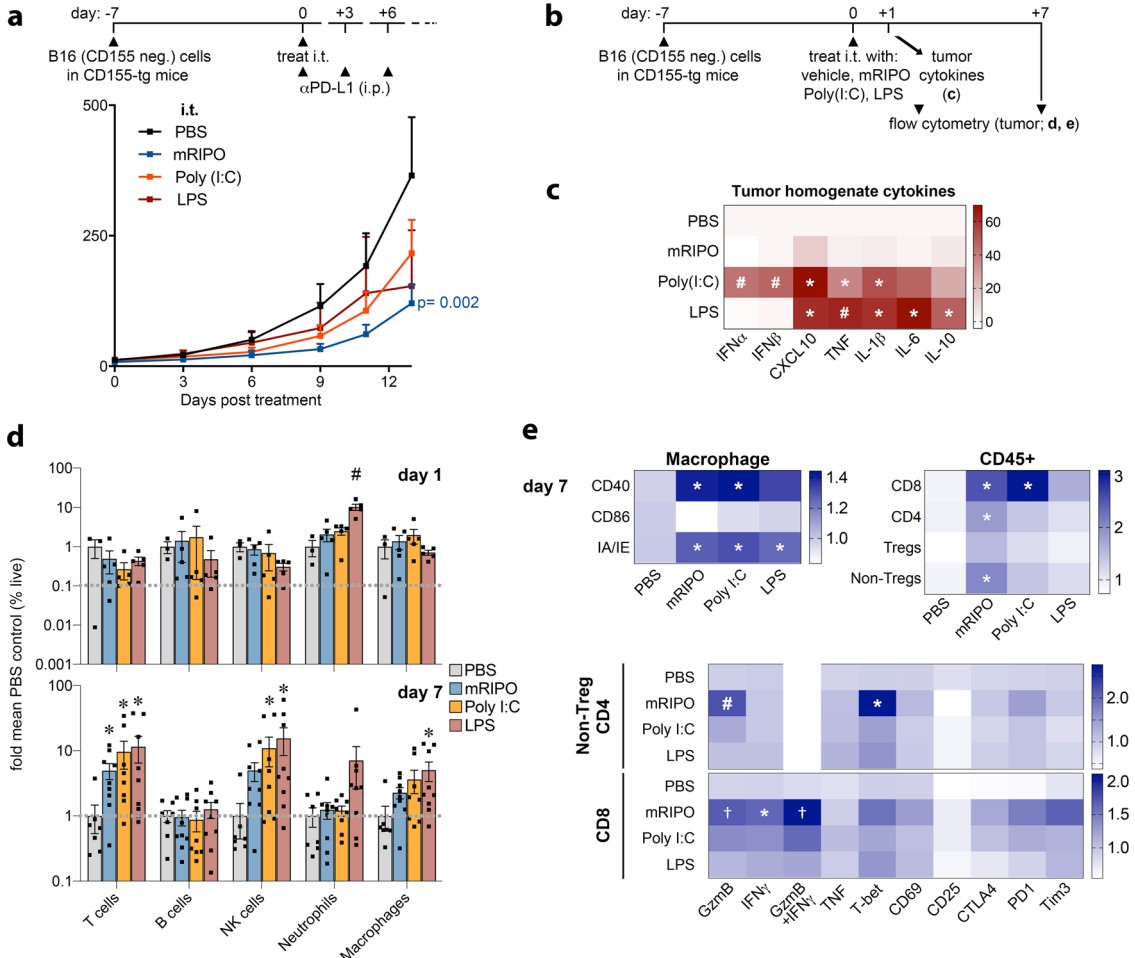

**Fig. 7 Virotherapy of the TME results in distinct TIL phenotypes. a** Wt B16.F10 tumors implanted in CD155-tg mice were treated with i.t. mock (PBS), mRIPO (10[7] pfu), LPS (30 µg), or Poly(I:C) (30 µg) with anti-PD-L1 therapy delivered i.p. to all groups as shown [$n = 9$ LPS, n = 8 for all other groups]; p-value is from two-way ANOVA vs mock. **b** CD155-tg mice bearing wt B16 tumors were treated with mRIPO ($3 \times 10^7$ pfu), Poly(I:C) (30 µg), or LPS (30 µg) for analyses in (**c–e**). **c** Tumor homogenate cytokines represented as %max for each cytokine followed by subtraction of mean PBS values ($n = 4$ PBS, $n = 5$ others) were measured 1 day after treatment. **d** Tumor immune cell subtypes as %live singlets at days 1 (top; $n = 3$ PBS, $n = 5$ others) and 7 [bottom; pooled from 2 experiments: $n = 7$ PBS, $n = 8$ Poly(I:C), $n = 9$ others]; mean −/+ SEM is shown. **e** TAM (n = same as d, day 7) or T cell activation markers for three experiments [$n = 9$ PBS, $n = 11$ mRIPO/Poly(I:C), $n = 12$ LPS]. Extended associated analyses and gating are shown in Supplementary Fig. 10; (**c–e**) asterisks denote two-tailed Tukey's post-hoc test $p < 0.05$ compared to (*) mock, (†) LPS, or (#) all groups.

type-I IFNs and the pivotal role of type-I IFNs in immune surveillance[34], we evaluated their contribution to T cell phenotypes. Poly(I:C) induced intratumor type-I IFN release, whereas LPS did not (Fig. 7c). Accordingly, combining Poly(I:C) with IFNα/β receptor (IFNAR)-blockade reduced the antitumor effect of Poly(I:C) (Fig. 9c). Moreover, supplementing intratumor LPS with recombinant IFNα achieved substantial antitumor efficacy (Fig. 9d), intratumor T cell influx, and reduced expression of PD1/TIM3 exhaustion markers on T cells (Fig. 9e; Supplementary Fig. 14a). IFNα monotherapy bolstered GzmB expression by CD8 T cells (Fig. 9e), but failed to mediate durable antitumor efficacy (Fig. 9d), infiltration of T-bet expressing CD4/8 T cells (Fig. 9e), or human myeloid cell activation (Fig. 4a; Supplementary Fig. 6d). Together, these results show that type-I IFN potentiates CD8 T cell function, but requires broader PRR-induced signaling to achieve durable antitumor immunotherapy.

**MDA5 signaling in the TME enhances antitumor efficacy.** Similar to PVSRIPO, transfected Poly(I:C) induced TBK1-IRF3 dominant innate signals in MDMs (Figs. 4, 5) and type-I IFN release in mouse PEC (Fig. 9f). If sustained TBK1-IRF3 signaling

promotes antitumor T cell function, then cytoplasmic delivery of Poly(I:C) should bolster intratumor T cell activity. Indeed, PEI-Poly(I:C) [analogous to transfected Poly(I:C)] yielded enhanced antitumor efficacy (Fig. 9g) and infiltration of GzmB/T-bet-expressing T cells (Supplementary Fig. 14b) compared to un-complexed Poly(I:C). PEI-Poly(I:C) is anticipated to engage both TLR3 (endosomal) and MDA5 (cytoplasmic). Since PVSRIPO-induced macrophage activation is largely MDA5-dependent (Fig. 6), we tested the specific role of MDA5 after PEI-Poly(I:C) treatment using MDA5 knockout (MDA5−/−) mice. MDA5 was not knocked-out in tumor cells and we did not test mRIPO due to the requirement for CD155-tg expression in mice. The antitumor efficacy of PEI-Poly(I:C) was impaired in MDA5−/− mice (Fig. 9h), and less T-bet and GzmB induction in TILs was observed (Fig. 9i). Together, these data reveal that cytoplasmic dsRNA selectively engages functional antitumor immunity and antitumor efficacy via MDA5 signaling in the TME.

**Discussion**

In this work, we delineate a virus:host relationship in the TME that achieves antitumor immunotherapy via sustained TBK1-

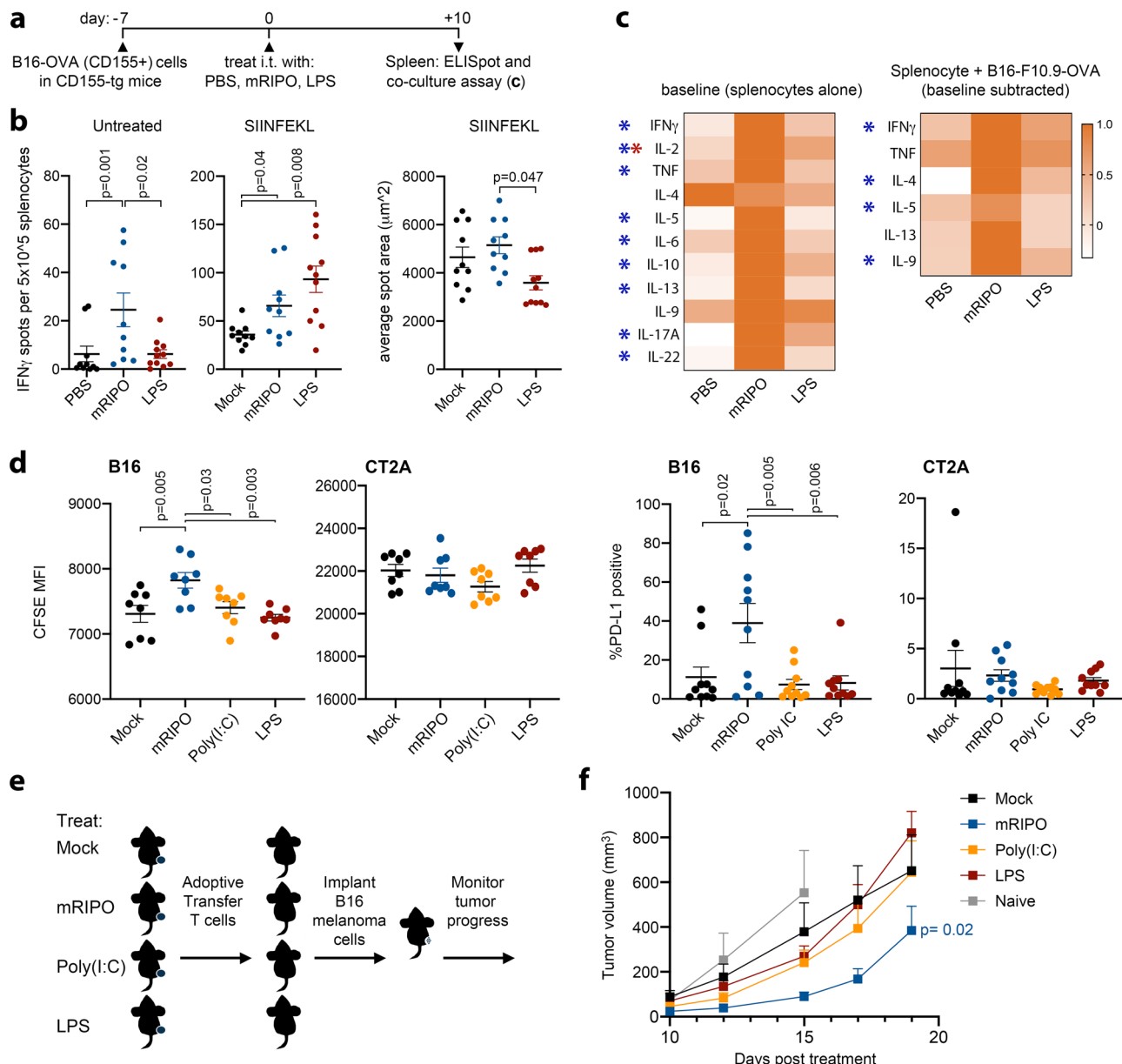

**Fig. 8 Virotherapy selectively induces functional antitumor T cell immunity. a** B16.F10.9[CD155]-OVA-implanted CD155-tg mice were treated as shown. **b** IFNγ ELISpot analysis of untreated splenocytes (left), or splenocytes cultured in the presence of SIINFEKL peptide (middle; untreated counts were subtracted). Right panel depicts average spot area ($n = 10$ for PBS/ mRIPO, $n = 11$ LPS pooled from 2 experiments); see Supplementary Fig. 11 for extended data. *P*-values are from Tukey's post-hoc (two-tailed) test. **c** Splenocytes from a subset of mice in (**b**) were tested in 48 h culture alone or with B16.F10.9-OVA cells (right). Supernatant cytokines are plotted for baseline (left) or antigen-specific secretion (right; baseline secretion was subtracted from B16. F10.9.OVA co-culture values). Heatmaps were normalized to maximum value for each cytokine ($n = 8$ mock/LPS; $n = 7$ mRIPO); only cytokines secreted above baseline are shown. (*) Tukey's post-hoc $p < 0.05$ (two-tailed), baseline mRIPO vs mock (blue asterisks): IFNγ $p = 0.03$, IL-2 $p = 0.004$, TNF $p = 0.03$, IL-5 $p = 0.01$, IL-6 $p = 0.02$, IL-10 $p = 0.04$, IL-13 $p = 0.008$, IL-17A $p = 0.01$, IL-22 $p = 0.004$; baseline LPS vs mock (red asterisk): IL-2 $p = 0.03$; baseline subtracted mRIPO vs mock (blue asterisks): IFNγ $p = 0.01$, IL-4 $p = 0.01$, IL-5 $p = 0.047$, IL-9 $p = 0.01$. **d** Tumor-draining lymph nodes (TDLNs, inguinal lymph node, $n = 8$ mice/group) from CD155-tg mice bearing wt B16 tumors treated with mock, mRIPO, Poly(I:C), or LPS as in Fig. 8a were harvested 7 days after intratumor treatment. TDLN cells ($5 \times 10^5$) were co-cultured with carboxyfluorescein succinimidyl ester (CFSE)-labeled B16 or CT2A cells in vitro ($3 \times 10^3$). CFSE median florescence intensity (left panels) and PD-L1 expression (right panels) were measured by flow cytometry for both cell lines after co-culture (48 h); see Supplementary Fig. 12 for gating and extended analyses; a representative series from three experiments is shown. *P*-values indicate Tukey's post-hoc two-tailed test. **e**, **f** Wt B16 tumor-bearing mice were treated as in (**d**) ($n = 5$/group). Ten days after treatment splenocytes were harvested and pooled by treatment. T cells were isolated by negative selection and adoptively transferred into naïve mice ($2 \times 10^6$ T cells per mouse) receiving B16 wt tumor cell implants on the same day; mean tumor volume + SEM is shown (**f**, $n = 8$ Mock/Poly(I:C), $n = 7$ mRIPO, $n = 6$ LPS, $n = 4$ naïve). *P*-value is from two-way ANOVA (two-tailed) comparing mice receiving T cells from mRIPO vs mock-treated mice; a representative series from two experiments is shown.

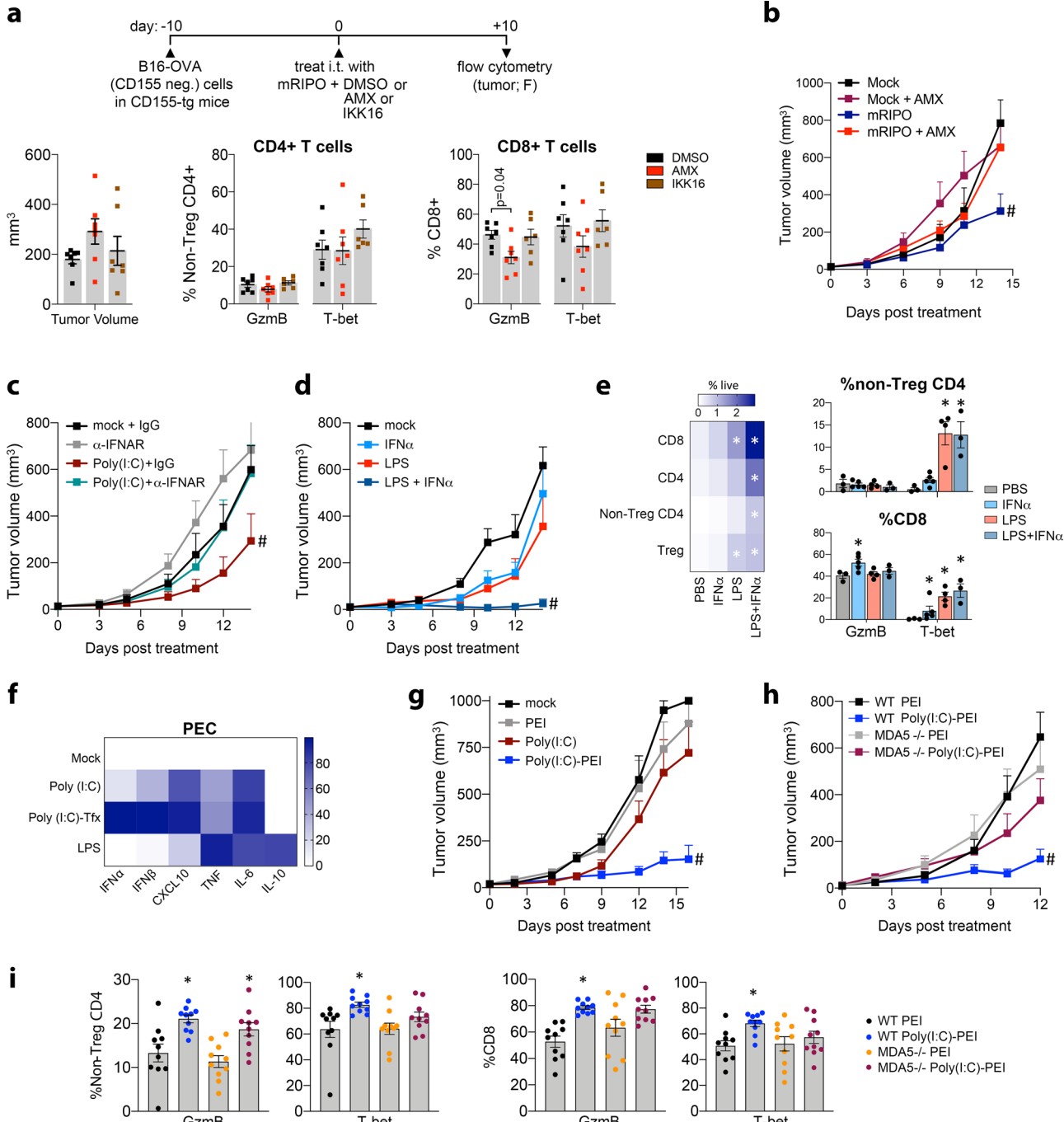

IRF3 innate signaling. Contrasting with TLR-mediated, IKKα/β-directed innate activation, intratumor TBK1-IRF3 polar inflammation selectively culminated in polyfunctional antitumor T cell immunity. Mechanistically, we define the role of TBK1-IRF3 controlled type-I/III IFN as a key determinant of T cell function within the TME. We conclude that virotherapy leverages sustained intratumor TBK1-IRF3 signaling to instigate type-I/III IFNs within a broader innate inflammatory context, which in concert, engages functional antitumor T cell immunity.

Our studies reveal an innate-stimulating mechanism of action for PVSRIPO via infection of TAMs/the TME. Such a scenario is consistent with radiographic responses in patients after intratumor PVSRIPO infusion, where evidence of acute inflammation abounds, and tumor regression, observed in a subset of patients, occurred over the course of months or years[5]; long after anticipated

viral persistence[7]. A trend of gradual tumor regression and intratumor immune inflammation was also documented with other cancer-targeting virus recombinants[35,36]. PVSRIPO, and PRR agonists, caused robust expression of immune-inhibitory/tolerizing molecules (i.e. PD-L1, IDO, A20, p50-NFκB, IL-10) in macrophages/DCs; thus, combination with immune checkpoint blockade, IDO inhibitors, and/or modalities that accentuate immune-activating macrophage/DC function (e.g. CD47 blockade or CD40 agonism) may potentiate antitumor effects of innate immune stimulants.

In line with the natural role for type-I IFN in driving antiviral CD8 T cell immunity[37], STING-mediated IFN responses induce spontaneous antitumor CD8 T cell immunity[38] and type-I IFN is critical to engage antitumor T cells[39,40]. Type-I IFN monotherapy has shown efficacy in some cancer types[41]. Consistent with its

**Fig. 9 MDA5-TBK1-IRF3 signaling within the TME mediates antitumor efficacy. a** (top) Treatment strategy combining mRIPO with DMSO, Amlexanox (AMX; TBK1:IKKε inhibitor, 250 nmol), or IKK16 (IKKα/β inhibitor, 140 nmol). **a** (bottom) Tumor volume at the time of harvest (left) and analysis of intratumor CD4+ (middle) and CD8+ (right) T cell expression of GzmB and T-bet ($n = 7$ DMSO/AMX, $n = 6$ IKK16; data bars represent mean $-/+$ SEM; Supplementary Fig. 13 presents extended data). $P$ value indicates Tukey's post-hoc two-tailed test. **b** Mice were treated with mock or mRIPO $-/+$ AMX (250 nm) as in (**a**) and mean tumor volume + SEM is shown ($n = 8$ mock + AMX, $n = 7$/group all others); (#) two-way ANOVA $p < 0.05$ vs all groups (two-tailed), mRIPO vs: mock $p = 0.0003$, mock + AMX $p = 0.01$, mRIPO + AMX $p = 0.01$. **c** Mock or Poly(I:C) (30 μg) was co-injected with 100 μg of isotype control IgG antibody or IFNAR-blocking antibody (all intratumor) into B16-F10-OVA tumor-bearing mice; mean tumor volume + SEM is shown ($n = 10$/group). (#) two-way ANOVA $p < 0.05$ vs all groups (two-tailed), Poly (I:C) + IgG vs: mock + IgG $p = 0.005$, mock + α-IFNAR $p < 0.001$, Poly (I:C) + α-IFNAR $p = 0.02$. **d, e** Mock or LPS was injected with and without 5000 IU of IFNα, mean tumor volume + SEM is shown (**d**, $n = 8$/group); (#) two-way ANOVA $p < 0.05$ vs all groups (two-tailed), LPS + IFNα vs mock and IFNα $p < 0.0001$, LPS $p = 0.002$. **e** Thirteen days after treatment, tumors were harvested from tumor-bearing, surviving mice for flow cytometry analysis ($n = 3$ mock/LPS + IFNα, $n = 5$ IFNα, $n = 4$ LPS only); heatmap (left) represents mean % live cells, data bars (right) represent mean $-/+$ SEM; see Supplementary Fig. 14a for extended analyses; (*) Tukey's post-hoc $p < 0.05$ (two-tailed) vs PBS control. **f** Mouse PEC were treated with mock, Poly(I:C) (10 μg/ml), Poly(I:C)-tfx (250 ng/ml Poly I:C-lyovec), or LPS (100 ng/ml); supernatant cytokines were measured, %maximum values were calculated, and mock values were subtracted from all other values; mean values are presented ($n = 2$ experiments). **g** Mice were treated i.t. with mock, PEI, Poly(I:C) (30 μg), or Poly(I:C) (8 μg) complexed with PEI [Poly(I:C)-PEI]. Tumor volume + SEM is shown ($n = 7$ PEI and Poly(I:C)-PEI, $n = 6$/group all others); see Supplementary Fig. 14b for associated flow cytometry analysis at day 16. (#) two-way ANOVA $p < 0.05$ vs all groups (two-tailed), Poly (I:C)-PEI vs all other treatment groups $p < 0.0001$. **h** WT or MDA5$-/-$ mice bearing B16 tumors were treated with PEI or Poly(I:C)-PEI as in (**g**). Tumor volume + SEM is shown ($n = 10$/group WT; $n = 11$/group MDA5$-/-$); (#) two-way ANOVA $p < 0.05$ vs all groups (two-tailed), MDA5 $-/-$ Poly (I:C)-PEI vs WT and MDA5$-/-$ PEI $p < 0.0001$, vs MDA5 $-/-$ PEI $p = 0.001$. **i** B16 wt tumors from mice treated as in (**h**, $n = 10$/group) were harvested eight days post-treatment and analyzed for T-bet and GzmB expression in tumor-infiltrating T cells, see Supplementary Fig. 14c for extended data; data bars depict mean $-/+$ SEM, (*) Tukey's post-hoc $p < 0.05$ (two-tailed) vs WT PEI; from left to right asterisks: $p = 0.004, 0.007, 0.02, 0.0006, 0.03$. All experiments were repeated at least twice and representative series are shown.

direct effects on CD8 T cells[42,43], we observed that recombinant IFNα increased GzmB and T-bet expression in CD8 T cells, but failed to elicit broader T cell/immune cell infiltration or durable antitumor efficacy. In contrast, we showed that when providing broader inflammatory context with LPS, recombinant IFNα mediated T cell-engaging antitumor efficacy. Similarly, despite mediating antitumor immunotherapy in a type-I IFN-dependent manner[44], intratumor therapy with Newcastle's Disease Virus induced T cell/myeloid cell recruitment, whereas recombinant IFNα did not[45]. Collectively, these findings underscore the importance of PRR-contextualized type-I IFN in engaging antitumor T cell immunity. A lack of such context may explain limited clinical efficacy of recombinant IFNα, despite the decisive roles for type-I IFN in immune surveillance. Indeed, the biological impact of type-I IFN is notoriously context-specific[46], e.g. during acute vs chronic viral infection[47].

PVSRIPO proficiently induces sustained TBK1-IRF3, type-I/III IFN-driven inflammation in human myeloid cells and primary human tumor tissues. Yet, mRIPO therapy evoked only modest pro-inflammatory cytokine induction in murine B16 tumors relative to LPS/Poly(I:C). Nonetheless, T cell polyfunctionality was higher after mRIPO treatment, in a TBK1-IRF3 dependent manner. Also, mimicking viral dsRNA by cytoplasmic delivery of Poly(I:C) induced prominent TBK1-IRF3 signaling and mediated superior antitumor immunotherapy relative to un-complexed Poly(I:C) in an MDA5-dependent manner. Our findings extend the role of sustained, MDA5-dependent, type-I IFN in determining functional CD8 T cell responses to LCMV infection[48]; they indicate the therapeutic potential of MDA5 signaling/RNA virus replication in the TME to efficiently stoke functional antitumor T cells. Since viral/dsRNA-induced MDA5 signaling led to pronounced IFN responses in the context of a broader inflammatory program (e.g. myeloid cell activation and immune cell influx), and IFN was critical for the functional status of TILs, we conclude that virotherapy elicits functional antitumor T cell immunity via PRR signals that encompass IFN. Our study identifies a cell-signaling nexus where PRR-specific signals (e.g. TLR4 vs. MDA5) impact distinct innate phenotypes and eventual T cell fate, in part, via tunable regulation of TBK1-IRF3 vs. IKKα/β-NFκB innate kinase assemblies.

Splenocytes and TDLN cells from mRIPO-treated mice secreted higher IFN-γ, IL-2, and other cytokines at baseline, indicating systemic bystander effects of intratumor viral infection/ replication. Such effects have been linked with type-I IFN responses in antigen-presenting cells after virus challenge[49,50]. Profuse splenocyte cytokine secretion was also observed, spanning Th1, Th2, and Th9-associated cytokines. Similarly, pathogenic poliovirus infection in mice induced both Th1 and Th2 cytokine secreting T cells (Th9 cytokines were not tested)[51]. Moreover, hybrid Th phenotypes, particularly Th1 and 2, are well documented in models of pathogen infection[52–54], including type-I/II IFN-dependent responses triggered by virus infection[55]. Therefore, we conclude that systemic changes in lymphocyte activity and phenotype mirror natural, anti-pathogen immune responses.

Altogether, our findings indicate distinct mechanisms by which polio virotherapy reinvigorates immune-surveillance and mediates antitumor efficacy. At the core, these depend on TBK1-IRF3 directed inflammation in the non-malignant TME. Thus, future studies into how TBK1-IRF3 signaling in the TME might be accentuated, sustained, or complemented are warranted.

## Methods

**Primary human tumor tissue and in vitro tissue slice culture assays.** Excess de-identified tumor tissue resected from patients as part of their clinical care was collected with patient consent under Duke University Institutional Review Board (IRB)-approved protocols. Pathologically confirmed GBM tumor tissue was collected within 1 h of resection by the Duke Preston Robert Tisch Brain Tumor Center BioRepository (PRTBTBR); de-identified breast cancer and melanoma tumor tissues (of varying grade and subtype) were collected by the Duke Biorepository and Precision Pathological Center (BRPC). For in vitro slice culture assay, tissue was sliced into quarters and subsequently sub-divided into ~2 mm diameter fragments using a #10 scalpel blade (Hill-Rom) and forceps. Slices derived from each quarter of the tumor were equally divided into 5 dishes (35 mm) containing 2 ml per well of medium [RPMI-1640 (Gibco); 10% FBS (Sigma-Aldrich); 100 ng/ml EGF (R&D Biosystems); 5 μg/ml Insulin (Gibco); 100 ng/ml hydrocortisone (Sigma-Aldrich)] generally following earlier procedures[56]. Samples were treated as shown in figure legends by adding virus/stimulants directly to cell culture media.

**Flow cytometry analysis of CD155 and phenotyping of slice cultures.** For analysis of cell type-specific CD155 expression in GBM tumor slices, tumor specimens were minced and dissociated in RPMI-1640 medium containing 100 μg/ml Liberase-TM (Sigma-Aldrich) and 10 μg/ml DNAse I (Roche) for 20 min at 37 °C

with agitation. Cell dispersions were filtered through 70 μM and 40 μM cell strainers (Olympus Plastics), washed in PBS (Gibco), and reconstituted in PBS containing 2% FBS (Sigma-Aldrich) with 1:20 Human Tru-Stain FcX block (BioLegend). Single-cell suspensions were stained with antibodies against CD45-BUV395 (BD Biosciences), CD14-BV421, CD33-BV510, CD49d-BV605, HLA-DR-BV786, CD31-FITC, CD3/19-BUV737, CD11b-APC, CD16-BV711, CD15-APC-fire7, and either CD155-PE or isotype control-PE antibody (all BioLegend); all at a 1:100 dilution. Viability stain (7-AAD, 1:100 dilution, BioLegend) was added after staining. For post-treatment analysis of tissue slices, slices were dissociated after treatment (48 h) as described above and Fc-blocked cell suspensions were stained with antibodies against CD45-BUV395 (BD Biosciences), CD14-BV421, CD31-FITC, CD11b-Alexa Fluor 488, CD16-BV711, CD15-APC-fire7, and 7-AAD, along with CD40-APC, CD86-BV510, PD-L1-BV605, and HLA-DR-BV786 or isotype controls for each fluorophore (all BioLegend at a 1:100 dilution). For tests of intracellular IFIT1, single-cell suspensions from tissue slices 48 h after treatment were stained with Zombie-Aqua viability dye (BioLegend) and fixed/permeabilized using the fixation/permeabilization buffer set (Thermo-Fisher) following manufacturer's instructions. Cells were Fc-blocked as above (1 h) followed by addition of IFIT1 antibody (Cell Signaling Technology; 1:100) or isotype control [purified rabbit polyclonal IgG (BioLegend); 1:100], and anti-dsRNA antibody (1:500, K1 clone, English and Scientific Consulting, Hungary) or normal mouse IgG (Santa-Cruz Biotechnology, 1:500) overnight. The next day, cells were washed twice with permeabilization buffer (Thermo-Fisher) and stained with donkey anti-rabbit IgG-Alexa Fluor 594 and goat anti-mouse IgG-APC (both BioLegend; diluted at 1:100) (1 h). Cells were then washed twice with permeabilization buffer and stained for antibodies recognizing CD45-BUV395 (BD Biosciences), CD14-BV421, CD31-FITC, CD11b-Alexafluor 488, and CD3/CD19-BUV737; all at a 1:100 dilution. K1 antibody staining did not show increased dsRNA signal after any treatment, including PVSRIPO infection, in any cell type or specimen despite separate validation of specific detection in PVSRIPO-infected HeLa R19 cells vs mock infected cells. PBMCs post-treatment with mock or PVSRIPO were similarly processed and analyzed. All cell suspensions were analyzed on a Fortessa X-20 cytometer (BD Biosciences). FCS files were analyzed using Flow-Jo version 10 (BD Biosciences). VersaComp beads (Beckmann-Coulter) and staining of fresh blood were used to standardize compensation.

**Cytokine analysis and cytokine T-SNE plots**. LEGENDplex (BioLegend) assays were used to measure cytokines with the Human/Mouse Antiviral, or Mouse Th Cytokine panels on a BD Fortessa X-20 flow cytometer. LEGENDplex software was used to assess analyte concentrations per manufacturer's instructions. Minimum threshold values were shown if cytokine concentrations were below sensitivity of detection (automatically calculated by analysis software); in rare cases where analyte concentration exceeded maximum threshold values, the maximum value was shown. Tumor homogenate for cytokine analysis was generated by immersing tumors in 1 ml PBS and mechanical disruption with a tissue homogenizer drill. T-SNE plots were generated from cytokine values (GBM slices: IL-1β, TNF, CXCL10, IFNλ1, IFNα, IFNβ, and IL-10; MDMs/ DCs: IL-1β, IL-6, TNF, IFNλ1, IFNα, IFNβ, IL-10, and IFNγ) using BioVinci software v1.1.5 (Bioturing, Inc).

**PBMCs, cell lines, and viruses**. Leukopaks (Stemcell Technologies) from 4 distinct de-identified donors were purchased after donors consented to an IRB approved protocol held by Stemcell Technologies. Leukopaks were processed using Leucosep tubes (Greiner Bio-One) and Ficoll-Paque Plus (GE healthcare) to isolate PBMCs following the manufacturer's instructions. B16-F10 cells (ATCC), E0771 cells (a gift from Greg Palmer, Duke University, USA), and B16-F10.9CD155-OVA cells[7] were grown in high-glucose DMEM (Gibco) supplemented with 10% FBS. B16-F10CD155 cells were derived by lentiviral transduction with CD155[7], followed by FACs purification of human CD155-expressing cells using CD155-PE antibody (BioLegend). Du54, U87, and Du43 cell lines[57] were grown in DMEM containing 10% FBS. All cell lines were confirmed to be mycoplasma negative (Duke Cell Culture Facility). PVSRIPO and CAV21 stocks were generated by propagation in HeLa R19 cells (E. Wimmer, State Univ. of NY, Stony Brook, USA) and isolation of supernatant 24 h after infection. Virus-containing supernatants were purified through a 0.1 μm syringe filter (Pall) followed by a 100Kda cutoff spin-filter (Millipore-Sigma) to remove small debris[57,58].

**In vitro PBMC experiments**. PBMCs were thawed, washed in 10 ml AIM-V media (Invitrogen), and incubated in 2 ml AIM-V media containing 10 μg/ml DNAse I (Roche) (15 min). Cells were spun down and plated at a density of $1 \times 10^6$ PBMCs per well in 6-well plates in DMEM (Gibco) containing 10% FBS, in the presence of 50 μg/ml M-CSF (Stemcell Technologies) for MDMs, or AIM-V media containing 50 μg/ml GM-CSF and 20 ng/ml IL4 for DCs, for 7 days. Media were changed (day 7) and assays were performed in the same culture vessel.

**Stimulants, inhibitors, and siRNA**. High molecular weight (HMW) Poly(I:C), LPS, 2′3′-cGAMP, HMW Poly(I:C) complexed with LyoVec transfection reagent, and Bx795 (all Invivogen); IKK16 and Amlexanox (Tocris Bioscience), BLZ945 (Selleckchem), IFNα2 and mouse IFNα3 (PBL Assay Science), were reconstituted per manufacturer's instructions; concentrations are noted in figure legends. SiRNA against MDA5 or All-Stars negative control siRNA (Qiagen) were complexed with lipofectamine 2000 (Thermo-Fisher) following manufacturer's instructions, delivering 100pmol of siRNA per 35 mm dish, and changing media after 6 h. Infections/treatments of siRNA-treated MDMs occurred 36 h after transfection.

**Immunoblots**. Immunoblots were performed using the Novex Protein Separation system with MES running buffer on 4–12% Bis-Tris gels (all Thermo-Fisher), 0.2 μm nitrocellulose membranes (G biosciences) for transfer, and combined magic mark XP and benchmark protein ladders (Thermo-Fisher) for molecular weight determination. Immunoblots were imaged using a Li-COR Odyessy Fc2 imaging system and Image Studio v5.2 (Li-COR) software for quantification. Antibodies against the following proteins were diluted in StartingBlock (Thermo-Fisher) except for phospho-specific antibodies, which were diluted in 5% bovine serum albumin (Sigma-Aldrich) in TBS-T (Cell Signaling Technology), at a 1:1,000 dilution unless otherwise indicated: p-IRF3(S396), IRF3, p-p65 NFκB(S536), p65 NFκB, IRF7, IRF1, P50/105 NFκB, P52/P100 NFκB, A20, c-REL, IκBα, IFIT1 (1:5,000), PKR, OAS1, p-STAT1(Y701), STAT1 (1:5,000), TAP1, p-p38, p38, PPARγ, Acetyl-coA Carboxylase (ACC), IDO, ISG15, MDA5, CD155 (Cell Signaling Technology); Tubulin (Sigma Aldrich; 1:10,000); poliovirus 2C (a gift of E. Wimmer, Stony Brook Univ., NY).

**Analysis of MDM/DC, PEC, and tumor-infiltrating immune cells**. MDMs and DCs were harvested from dishes by incubation in 0.25% Trypsin-EDTA (Thermo-Fisher; 5 min), followed by addition of growth media and centrifugation. Cells were resuspended in PBS containing 2% FBS with Fc block (1:50 Tru-stain FC block, BioLegend) and incubated at RT (30 min). The following antibodies or matched isotype controls were used at a dilution of 1:250 for staining: CD40-APC, CD86-BV510, CD83-BV711, PD-L1-BV605, HLA-ABC-FITC, HLA-DR-BV786, and CCR7-PE (all BioLegend). For analysis of PEC phenotype, post-infection staining was performed for CD40-BV605, CD86-PEcy7, PDL1-APC, I-A/I-E (MHC-class II)-BV786, H2Kb (MHC-class I)-PE and 7-AAD (all BioLegend at a 1:250 dilution). PEC and BMDC cultures were separately stained for CD45-BUV395, CD11b-BV711, and CD11c-APC (all 1:250) to confirm purity of macrophages or DCs, respectively. For analysis of tumor-infiltrating cell density/phenotypes on tumor-derived, single-cell suspensions (dissociated as described for human tumors) multiple antibody staining panels were used, diluting all antibodies 1:100 unless otherwise specified: lineage [Zombie-Aqua (1:250), CD45-BUV395, CD11b-BV711, F4/80-PECy5, Ly6G-PE, NK1.1-BV421, CD3-FITC, CD19-FITC, CD40-BV605, CD86-PECy7, I-A/I-E-BV786; isotype control stained samples for CD40, CD86, and I-A/I-E were performed], T cell activation/ exhaustion [Zombie-Aqua (1:250), CD45-BUV395, CD3-APC, CD4-PE, CD8-BV421, CD69-BV605, CD44-PECy5, CD62L-BV786, TIM3-BV711, and PD1-PECy7], Treg staining was performed using the FoxP3 staining kit (Invitrogen) following manufacturer's instructions with additional staining antibodies [Zombie-Aqua (1:250), CTLA4-BV605 (intracellular), CD45-BUV395, CD3-APC, CD4-FITC, CD8-BV421, FoxP3-PECy5 (1:50), CD25-PE, and PD1-PECy7; isotype control for FoxP3 was performed], and intracellular T cell staining panel [Zombie-Aqua (1:250), CD45-BUV395, CD3-APC, CD4-FITC, CD8-BV421, FoxP3-PECy5 (1:50), Granzyme B-PECy7 (1:50), IFNγ-BV786 (1:50), TNF-BV605 (1:50), and Tbet-BV711 (1:50); isotype controls for Granzyme B, IFNγ, TNF, and T-bet were performed]. All flow cytometry antibodies/reagents were obtained from BioLegend, with exception of CD45-BUV395 (BD Biosciences), FoxP3, CD4, and CD25 (Thermo-Fisher). Gating strategies are shown in supplementary figures; isotype controls, fluorescence minus one controls, and staining of blood were used to assess positive stained populations.

**Cell separation assays**. CD14+ cells from GBM single-cell suspensions/PBMCs were isolated using the human CD14+ selection kit II (Stemcell Technologies). For GBM single-cell suspensions one-third of the sample was retained as pre-depletion sample, the remaining material was routed through CD14+ selection. Pre- and post-depleted cells were adjusted for cell counts with equal viable cell density using a Countess II automated cell counter with trypan blue staining (Thermo-Fisher; CD14 cell density was not normalized to match density of pre- and post-depletion samples). Samples were plated in 24-well plates, and treated with: mock solution (DMEM), PVSRIPO (MOI 10), Poly(I:C) (10 μg/ml), LPS (100 ng/ml), or 2′3′-cGAMP (5 μg/ml) (48 h). Depletion of CD14 cells was confirmed for each specimen by flow cytometry using antibodies against CD45-BUV395, CD14-BV421, and CD11b-APC (all 1:100 dilution).

**Mice**. C57BL/6J mice (strain code 000664) and MDA5 −/− C57BL6J (strain code 015812)[27] were purchased from Jackson Labs. CD155-tg C57BL/6 mice were a generous gift of Satoshi Koike (Tokyo Metropolitan Institute of Medical Science, Japan). Six- to ten-week-old male and female mice were used for B16 studies, with roughly equal distribution of males and females in each treatment group; female mice were used for E0771 studies (8–10 weeks old) and MDA5−/− vs WT comparison studies (6–8 weeks old) in the B16 model. Mice were housed in the Duke University Cancer Center Isolation barrier facility with 12-hour light/dark

cycles, relative humidity of 50 −/+ 20%, and temperature of 21 −/+ 3 °C. All mice were used in accordance with Duke IACUC-approved protocols.

**Mouse tumor model experiments**. The flanks of WT or CD155-tg C57BL6/J mice were injected subcutaneously with 50 µl of $2.5 \times 10^5$ B16-F10, B16-F10$^{CD155}$, or B16-F10.9$^{CD155}$-OVA cell suspensions in PBS. E0771 cells were implanted into the 4th mammary fat pad at a density of $5 \times 10^5$ cells in 50 µl of PBS. Tumor volume was measured using vernier calipers (Thermo-Fisher) and calculated using the equation: volume = L x W x W/2. Mice were euthanized via $CO_2$ inhalation upon reaching a total tumor volume of 1000 mm3 or ulceration of the tumor; which was the endpoint in all tumor model experiments. For PEC co-implantation experiments, 0.2 µM-filtered thioglycolate (Sigma-Aldrich) elicited PEC were harvested from CD155-tg mice[59] and plated at a density of $1 \times 10^7$ cells in 100 mm$^3$ cell culture dishes in DMEM containing 10% FBS. After overnight adherence, media was changed and cells were treated with mock (DMEM) or mRIPO ($1 \times 10^8$ pfu) (24 h). PECs ($1 \times 10^5$) were then trypsinized, washed, and mixed 1:2.5 with B16 cells, or 1:5 with E0771 cells, followed by implantation as described above. A portion of the cell mixes was plated in 96 well plates ($3 \times 10^4$ cells per well) for WST1 (Sigma-Aldrich) tests per manufacturer's instructions using a ChroMate spectrophotometer during a 3-day time course. For virus co-implantation experiments using E0771, tumor cells were co-implanted with the addition of mock (DMEM) or mRIPO ($1 \times 10^7$ pfu). For the generation of BMDCs, isolated bone marrow cells from femurs and tibias were cultured at a density of $2.5 \times 10^6$ cells/ml in RPMI + 10% FBS containing 300 ng/ml human recombinant FLT3L (Thermo-Fisher) for 9 days and non-adherent cells were used[8]. FLT3L-derived BMDCs were tested for CD11c expression by flow cytometry (CD11c-APC at 1:100, Biolegend). For PEC and BMDC therapy assays, B16-F10 cells were implanted in the flanks of C57BL/6 J mice, followed by injection of mock- or mRIPO-infected (MOI 10, 24 h) PEC ($1 \times 10^6$ cells per tumor) or BMDCs ($5 \times 10^5$ cells per tumor) into tumors at day 7 in a volume of 50 µL. For mRIPO and LPS therapy experiments B16-F10, B16-F10$^{CD155}$, or B16-F10.9$^{CD155}$-OVA tumors were injected 8 days after implantation with PBS, mRIPO ($1 \times 10^7$ pfu), or LPS (30 µg). PD-L1 blocking antibody (BioXcell) was used at a dose of 250 µg per mouse injected intraperitoneally. IFNAR-blocking antibody (100 µg; BioXcell) or control was mixed with water or Poly(I:C) for intratumor injection. For co-treatment of mRIPO and TBK1 (AMX) or IKKα/β (IKK16) inhibitors, virus preparations were combined with inhibitors with equal concentrations of DMSO and injected directly into tumors. In vivo-jetPEI (Polyplus transfection) was used for injection of mouse tumors following manufacturer's recommendations: 1.2 µl of PEI reagent + 8 µg of Poly(I:C) was used per mouse; control mice were treated with 1.2 µl of PEI reagent + physiological water (Invivogen). All cell counts were performed using Countess II (Thermo-Fisher); tumor measurements were performed blinded from treatment group allocation.

**ELISpot, splenocyte/TDLN co-culture assays, and adoptive transfer**. Mice bearing B16-F10.9$^{CD155}$-OVA tumors treated with PBS, mRIPO, or LPS as described above (without anti-PD-L1 blockade) were euthanized 10 days after treatment for harvesting and processing their spleens[8]. ELISpot assays were performed blinded to treatment group allocation in randomized order and read by Zellnet Consulting, Inc.[8]. Samples with the two highest and two lowest IFNγ spot counts after SIINFEKL stimulation were excluded from each group to remove outliers. For splenocyte co-culture assays, B16-F10 or B16-F10.9-OVA cells were plated in 96 well plates ($5 \times 10^3$ cells per well) the day before spleen harvest. Cell-count normalized splenocytes remaining after ELISpot assay preparation were added at a density of $1 \times 10^6$ splenocytes per well with no tumor cells, or B16-F10.9-OVA tumor cells and cultured in RPMI-1640, 10% FBS (48 h). The mouse Th Cytokine Panel LEGENDplex (Biolegend) was used to analyze supernatant cytokines. The highest and lowest values for each cytokine were removed from each group to remove outliers. For TDLN co-culture assays TDLNs were harvested seven days after treatment, crushed, and counted. Separately B16 or CT2A cells were stained with the CFSE cell division tracker kit (BioLegend) following the manufacturer's instructions. B16 or CT2A cells ($3 \times 10^3$) were plated in separate wells of 96 well plates and TLDN cells ($5 \times 10^5$) were added. Forty-eight hours later, cells were stained (CD45.2-BUV395, CD3-APC, CD8-BV421, CD4-PE, OX40-BV711, PD-L1-BV605, and 7-AAD; all 1:250) and analyzed by flow cytometry. For adoptive transfer experiments splenocytes were pooled by treatment group, T cells were isolated using the EasySep Mouse T cell isolation kit (Stemcell Tech, negative selection), and isolation of T cells was confirmed by flow cytometry. Two million T cells in PBS were transferred by I.P. injection immediately following implantation of B16 cells ($2.5 \times 10^5$) in naïve recipients.

**Statistical analysis**. Assay-specific statistical tests are indicated in the corresponding figure legends. GraphPad Prism v8 was used to perform all statistical analyses and plot data. A statistical probability of <0.05 ($p < 0.05$, two-tailed) was used unless otherwise noted in the figure legend (i.e. when Bonferroni correction was used). Unless provided in the figure or figure legends, exact p-values are provided in the source data file for relevant statistical comparisons. All data points reflect individual specimens, independent experimental repeats, or mice.

**Reporting summary**. Further information on research design is available in the Nature Research Reporting Summary linked to this article.

## Data availability
All data are available in the article and supplementary information files or from the corresponding author upon reasonable request. Source data are provided with this paper.

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

## Acknowledgements

The authors acknowledge the excellent support with patient tissue procurement provided by the Preston Robert Tisch Brain Tumor Center Biorepository and the Duke Precision Bio-Repository and Pathological Center. We thank V. Chandramohan and D. Boczkowski for insightful discussions and technical advice, G. Palmer (Duke University, NC) for E0771 cell line, and Satoshi Koike (Tokyo Metropolitan Institute of Medical Science, Japan) for CD155-tg C57BL6 mice. Funding sources; PHS: F32CA224593 (M.C.B.), R01NS108773 (M.G. and S.K.N.), R35CA225622 (D.D.B.); Department of Defense Breast Cancer Research Program award W81XWH-16-1-0354 (S.K.N.) and National Cancer Center Breast Cancer Project Grant (M.C.B.).

## Author contributions

M.C.B. contributed to the conception, study design, acquired data, analyzed data, and wrote the manuscript; M.M.M. acquired data; M.M. contributed to study design; Z.P.M. acquired data; E.K.H. acquired data and provided technical expertise; J.P.K contributed to study design; Y.Y. acquired data; G.M.B. contributed to study design and provided reagents; E.S.H contributed to study design and provided reagents; D.M.A. contributed to study design; D.D.B contributed to study design and provided reagents; S.K.N. contributed to the conception, study design, and manuscript writing; M.G. contributed to the conception, study design, analysis of data, and manuscript writing. All authors participated in reviewing and editing.

## Competing interests

M.C.B., D.D.B., D.M.A., S.K.N., and M.G. own intellectual property related to this research, which has been licensed to Istari Oncology, Inc. M.G. and D.D.B. are compensated advisors to- and own equity in Istari Oncology, Inc. S.K.N., M.C.B., D.D.B., and M.G. are inventors on patent application PCT/US2017/039953 held/submitted by Duke University that covers the composition and methods for activating antigen-presenting cells with PVSRIPO. All other authors declare no competing interests.
