## [Peer Review File · Nature Communications]

REVIEWER COMMENTS

Reviewer #1 (oncolytic viruses) (Remarks to the Author):

This manuscript aims to define immunological mechanisms of action of a recombinant chimeric oncolytic poliovirus independent of overt direct oncolysis. The conclusions are that the virus directly infects non-malignant cells inside tumors, preferentially induced innate immune responses via the TBK1-IRF3 signaling axis and this was associated with robust and sustained type I interferon responses. In turn, polyfunctional Th1-associated T cells were induced. This could not be recapitulated by a panel of ligands for pattern recognition receptors on their own.

Experiments were generally comprehensive, sophisticated and appropriate. A large amount of data were generated and the main text was easy to follow.

Minor Issues:

Page 3: "thereby arousing the immune system in the context it evolved to function: pathogen infection/replication". This is speculative and does not account for the immune system's development of autoimmune effector cells to respond to and eliminate dangerous aspects of self, including cancer cells.

All abbreviations should be defined (e.g. "PRR" on page 3). The manuscript should be checked carefully for this.

Page 1, phase-1 testing: What is the comparator? ...what was the % long-term survival in controls? This should be stated. Otherwise, the statement has no context.

All abbreviations should be defined in the figure legends.

Fig. 1b: No statistically significant differences are shown. Were there any differences? The color scheme for "m'phages" vs. "neutrophils" was difficult to distinguish in the legend (the colors are too similar).

Fig. S1: the text in the dot plots is too small. (A) Which leukocyte subsets are represented by each phenotype (the reader is forced to look these up)? What sequence is the reader to follow through the multiple panels? Are controls based on 'fluorescence minus one' or isotype controls?

Figure 1c is being used to argue that there is limited oncolysis. Can a positive control be provided to demonstrate that the virus was replication-competent and of a sufficient dose to induce oncolysis in susceptible cells? Was only one experimental replicate performed? Why was only one slice from a breast cancer used? ...what is the biological significance of a n=1?

Fig. 1e: The legend suggests there will be data for four treatments but it appears to show data for only one. Which treatment is shown?

Fig. 1: Why weren't samples dilute to assess IL-6 (i.e. to bring it within the linear range of detection). The fact that it was 'over-produced' suggests it may have been a dominant cytokine and may be very important for the biology being assessed.

Fig. 1f: Were other dark purple/blue squares different from mock (e.g. CD40 mock vs. PVSRIPO)?

Fig. 1h: Does the asterisk apply to "PVSRIPO" or one of the bars

Fig. 2a: What do blue vs. red asterisks indicate? The legend defines a black asterisk.

Material and Methods: What was the sex and age of the mice that were used in the studies? What equipment was used to make tissue slices? How long did tissue slices remain viable?

There were too many additional minor issues to document in a timely fashion. The manuscript (especially figure and supplementary figure legends) should be revised to add clarity for readers.

Major Issues:

There were a lot of references to functionality of T cells throughout the manuscript. However, the assessments determined complex phenotypes that are often associated with improved functions. Direct functional assessments should be performed (e.g. co-culture cytotoxicity assay, depletion of T cells in efficacy studies). This would also help determine whether the T cells that are being assessed are tumor-specific or virus-specific (or have some other specificity).

The authors highlight that the findings are independent of 'overt' oncolysis. However, in the absence of positive controls, data to argue against oncolysis in the manuscript are weak. Regardless, why was a model with relatively weak oncolysis used? What would the results be if the virus were studied in a model in which robust oncolysis occurs? In contrast, what would the results look like if the virus were to be irradiated (rendered replication incompetent). A comparison between the virus and ligands for pattern recognition receptors seems irrelevant if the virus can utilize oncolysis (even if it is limited) to stimulate immune responses by other mechanisms.

The method used to culture murine DCs was not described and this is very important with respect to drawing conclusions regarding DC biology. If not sorted, standard "DC cultures" are actually dominated by macrophages and other cells (see: <https://doi.org/10.1016/j.immuni.2015.05.018>). The authors need to prove that they were studying DCs.

Conclusions were drawn regarding infected cells (e.g. macrophages and DCs) but no data were provided to show that these cells were infected. They were treated with the virus but many not have been infected.

A mechanism of action for oncolytic poliovirus was proposed. How do the authors envision harnessing such knowledge to improve virotherapies for cancers?

Reviewer #2 (oncolytic viral vectors) (Remarks to the Author):

What are the major claims of the paper?

- That recombinant poliovirus RIPO therapy primarily mediates antitumor immunotherapy via direct infection of the non-malignant tumor microenvironment, independent of overt malignant cell lysis.
- Relative to other innate immune agonists, virotherapy provoked selective, exaggerated TBK1-IRF3 driven innate inflammation that was associated with sustained type-I/III interferon (IFN) release.
- Despite priming equivalent antitumor T cell quantities, virus-mediated TBK1-IRF3 signaling, but not NFκB-polarized TLR activation, culminated in polyfunctional and Th1-differentiated antitumor T cell phenotypes.
- Recombinant type-I IFN bolstered tumor-localized T cell function but did not mediate durable antitumor immunotherapy without concomitant pattern recognition receptor (PRR) signaling. Thus, virus-induced TBK1-IRF3 signaling in the TME provides PRR-contextualized IFN responses that elicit functional antitumor T cell immunity.
- TBK1-IRF3 innate signal transduction determines eventual function and differentiation of tumor infiltrating T cells.

Are the claims novel? If not, please identify the major papers that compromise novelty.

The novelty of the manuscript is limited . For example, "STING senses the presence of nucleic acids from intracellular pathogen infection and in turn initiates a downstream signaling cascade that includes TBK1-mediated activation of IRF3 and Stat6, resulting in IFN-I and cytokine production." J Cell Commun Signal. 2018 Mar; 12(1): 83–90.

Evaluating TBK1 as a Therapeutic Target in Cancers with Activated IRF3 DOI: 10.1158/1541-7786.MCR-13-0642 Published July 2014.

Will the paper be of interest to others in the field?
Yes.

Will the paper influence thinking in the field?

Not markedly. Aside from modest novelty, another reason is that effects seen in Figure 2 are modest. Infection with mRIPO of PEC, BDMC or non-malignant TME has modest, though significant, anti-tumor effects. Likewise, for Figure 6A. Figure 7C shows that the most efficacious effect is seen by Poly(I:C)-PEI. In 7E LPS + IFN α is most efficacious.

Are the claims convincing? If not, what further evidence is needed?

The data provided is of high-quality and credible. The major problems remain novelty and efficacy.

Are there other experiments that would strengthen the paper further? How much would they improve it, and how difficult are they likely to be?

Additional experimentation might provide an experimental protocol that might be more efficacious and enhance the potential of the suggested approach using viral infection of the tumor microenvironment via selective Tbk1-Irf3 signaling. These are no likely to be easily accomplished in a short period of time.

Are the claims appropriately discussed in the context of previous literature?

There is discussion of the literature, though the novelty of the approach give studies by other groups might be overstated.

If the manuscript is unacceptable in its present form, does the study seem sufficiently promising that the authors should be encouraged to consider a resubmission in the future?

The data are sound, and the manuscript is well written. The manuscript appears to have a modest impact. Not certain that this impact can be markedly enhanced in the near term.

Is the manuscript clearly written? If not, how could it be made more accessible?

Could the manuscript be shortened to aid communication of the most important findings?

Have the authors done themselves justice without overselling their claims?

The manuscript is well written, but impact is overstated.

Have they been fair in their treatment of previous literature?

Greater discussion of related findings by others would provide a more comprehensive background for the reader.

Have they provided sufficient methodological detail that the experiments could be reproduced?

Yes.

Is the statistical analysis of the data sound?

Yes

Should the authors be asked to provide further data or methodological information to help others replicate their work? (Such data might include source code for modelling studies, detailed protocols or mathematical derivations).

Not necessary.

Are there any special ethical concerns arising from the use of animals or human subjects?

There are no apparent concerns.

Reviewer #3 (tumour microenvironment) (Remarks to the Author):

The ms by Brown et al investigates virotherapy by a recombinant poliovirus (PVSRIPO) that shows promise in clinical trials as a means to combat solid tumors via stimulating tumor immunity. This study reports that improved antitumor immunity using this method is driven by innate immune

responses by non-malignant TME cells (including a strong response by TAMs) and not lysis of tumor cells themselves. This was seen for GBM, breast and melanoma indicating a tumor type independent response. Further, this response is characterized by an MDA5-TBK1-IRF3 mediated IFN response, that when pharmacologically inhibited, reduced the efficacy of this virotherapy in mice. Of interest, IFNs or PAMPs alone were not sufficient to drive this response indicating a distinct signal sent by the live virus during its replication. Further, recombinant poliovirus therapy triggered a robust T cell response that was characterized by the expansion of polyfunctional effector T cells.

This is an interesting study and a well written paper that makes use of primary tumor sections and a newly established mouse model of PVSRIPO infection (mRIPO, CD155 Tg mice) to interrogate the mechanistic basis for this specific form of virotherapy against solid tumors. The experiments are well done and data clearly presented, and the work collectively provides important insights into how PVSRIPO is able to bolster antitumor immunity.

Despite my enthusiasm, some minor weaknesses include the following:

1. Models whereby MDA5, TBK1 and/or IRF3 are genetically lacking from TAMs and other cell types within the TME would provide stronger mechanistic support for the conclusions. However, the TBK1 inhibitor and IFNAR blocking Ab experiments do provide some functional data that support their proposed model of PVSRIPO antitumor function, and should suffice. While the TBK1 inhibitor (AMX) did reduce GzmB CD8+ T cells, tumor growth data from this experiment should also be shown, as it would be expected that tumor growth would increase.
2. It would strengthen the argument that heightened and sustained TBK1-IRF3 is the critical driver of the antitumor response mediated by PVSRIPO if the authors could devise a gain of function approach to specifically increase and sustain this pathway in the TME.
3. Can the authors elaborate further on how they think PVSRIPO is able to drive polyfunctional T cell phenotypes, and why the PAMPs did not?
4. Statistical significance is not demonstrated for some figures, and this makes it difficult to determine if some differences, such as differences in tumor growth in figures 2, 6 and 7, are meaningful. This should be added. Further, number of experimental replicates should be noted in the legend for each figure.

Reviewer #1 (oncolytic viruses) (Remarks to the Author):

This manuscript aims to define immunological mechanisms of action of a recombinant chimeric oncolytic poliovirus independent of overt direct oncolysis. The conclusions are that the virus directly infects non-malignant cells inside tumors, preferentially induced innate immune responses via the TBK1-IRF3 signaling axis and this was associated with robust and sustained type I interferon responses. In turn, polyfunctional Th1-associated T cells were induced. This could not be recapitulated by a panel of ligands for pattern recognition receptors on their own.

Experiments were generally comprehensive, sophisticated and appropriate. A large amount of data were generated and the main text was easy to follow.

Minor Issues:

1. Page 3: “thereby arousing the immune system in the context it evolved to function: pathogen infection/replication”. This is speculative and does not account for the immune system’s development of autoimmune effector cells to respond to and eliminate dangerous aspects of self, including cancer cells.

The reviewer is right, and this speculative statement was removed. The sentence was revised as follows: ‘Acute viral challenge of malignant tumors, e.g. via recombinant viruses, may achieve both cancer-cell killing and innate immune activation, thereby arousing adaptive antitumor immunity’ (**revised manuscript, pg. 3**).

2. All abbreviations should be defined (e.g. “PRR” on page 3). The manuscript should be checked carefully for this.

Done.

3. Page 1, phase-1 testing: What is the comparator? ...what was the % long-term survival in controls? This should be stated. Otherwise, the statement has no context.

This information has been added, pasted below for the reviewer’s convenience:
‘Phase-1 testing of intratumoral PVSRIPO, a highly attenuated rhino:poliovirus chimera², revealed durable radiographic responses and a 21% survival rate in recurrent glioblastoma (GBM) patients at 36 months post therapy, relative to 4% survival of a criteria-matched historical control cohort³’ (**revised manuscript, pg. 3**).

4. All abbreviations should be defined in the figure legends.

Done.

5. Fig. 1b: No statistically significant differences are shown. Were there any differences? The color scheme for “m’phages” vs. “neutrophils” was difficult to distinguish in the legend (the colors are too similar).

We have added the requisite statistics to Figure 1b (**revised Figure 1b**). All cell types, except for neoplastic cells, significantly express CD155 as compared to no expression (normalized MFI=0, defined by subtracting isotype control MFI) using a Wilcoxon rank sum test (**revised**

Figure 1b). We fixed the problem with the coloring, which was indeed difficult to decipher (**revised Figure 1b**).

6. *Fig. S1: the text in the dot plots is too small.*

The reviewer is correct. We have enlarged all displays, esp. for the gating strategies presented in the Supplement, so that the fine print legends are clearly legible.

7. *Which leukocyte subsets are represented by each phenotype (the reader is forced to look these up)?*

We have added cell type names (**revised Supplementary Figure 1**).

8. *What sequence is the reader to follow through the multiple panels?*

We have added the origin gate on top of each downstream gate (**revised Supplementary Figure 1**).

9. *Are controls based on 'fluorescence minus one' or isotype controls?*

Isotype control-stained samples that were stained with all other antibodies (fluorescence minus one isotype control) were used as the control. This information was added to the figure legend (**revised Supplementary Figure 1 legend**), and is included in the description of the summary data shown in Fig 1b.

10. *Figure 1c is being used to argue that there is limited oncolysis. Can a positive control be provided to demonstrate that the virus was replication-competent and of a sufficient dose to induce oncolysis in susceptible cells?*

Also see our response to Editorial comment nr. 3 (pg. 3, above). Viability data from positive control GBM cell lines color (Du54, U87 and Du43) infected with PVSRIPO (using the same dose of virus) are now included, shown in red color (**revised Figure 1c**). We also confirmed that in tissue slices, 7-AAD effectively and accurately detected dead cells by incubating tissue slice samples for 20min at 95°C prior to 7-AAD addition and flow cytometry. This confirmation is shown below in **Figure R1**:

Figure R1. 7-AAD effectively detects cell death in tissue slices.

11. Was only one experimental replicate performed? Why was only one slice from a breast cancer used? ...what is the biological significance of a n=1?

We included only a single breast cancer specimen in our analyses because sufficient single cell-suspension material was required to do the viability/macrophage activation assessment. Two breast cancer specimens provided enough cell suspension material for flow cytometry analysis (typically, breast cancer specimens contain mostly fat and stroma), and only one was tested for viability with 7-AAD due to prioritization for testing of IFIT1 expression (in Figure 1g).

It is correct that 1 specimen is insufficient to generalize about the extent of oncolysis in breast cancer tissue slices. Therefore, we have removed this specimen from Figure 1c, which now only includes data from GBM specimens (**revised Figure 1c**). We removed the breast cancer specimen in Figure 1f as well, for the same reason (now, six GBM specimens are included; **revised Figure 1f**). We retained the IFIT1 expression data including two breast cancer specimens tested, which were grouped with 4 GBM specimens, as the focus was tumor-associated macrophages in general, and not breast cancer- compared to GBM neoplastic cells proper.

12. Fig. 1e: The legend suggests there will be data for four treatments but it appears to show data for only one. Which treatment is shown?

The reviewer is correct about this oversight: we apologize for the confusion. The figure legend has been revised; only PVSRIPO was tested (**revised manuscript, pg. 25**). For both breast cancer and melanoma specimens, only limited tissue was available for our analyses. This precluded testing of PRR agonists other than PVSRIPO, since the primary purpose of our investigations was to define the treatment response in intact tumor tissue to PVSRIPO (PVSRIPO is currently in clinical trials in glioblastoma, melanoma and breast cancer).

13. Fig. 1: Why weren't samples dilute to assess IL-6 (i.e. to bring it within the linear range of detection). The fact that it was 'over-produced' suggests it may have been a dominant cytokine and may be very important for the biology being assessed.

The reviewer is correct. We have re-analyzed supernatant specimens where IL-6 was originally excluded due to the above-mentioned limit-of-detection issues (**revised Figure 1e**). This did not change the interpretation of our data, but allows for a more confident assertion that PVSRIPO primarily induces type-I/III IFNs and CXCL10 in treated tumor tissues, while TLR agonists dominantly induce TNF- α , IL-6 and IL-1 β .

14. Fig. 1f: Were other dark purple/blue squares different from mock (e.g. CD40 mock vs. PVSRIPO)?

The differences that were observed were not statistically significant, though CD40 induction was observed in most cases after PVSRIPO treatment. To account for this, we have moved the CD40 and HLA-DR data to the Supplement, where individual values are included and the raw data can be evaluated in detail (**new Supplementary Figure 3b**). Interestingly, despite robust

activation marker expression in monocyte-derived macrophages after treatment with PVSRIPO and other PRR agonists, tumor-associated myeloid cells primarily induced PD-L1 in response to innate stimuli/PVSRIPO (**new Supplementary Figure 3b; Figure 1f**).

15. Fig. 1h: Does the asterisk apply to “PVSRIPO” or one of the bars

The asterisk applies to the sum log-fold mock control values for all cytokines comparing pre- vs. post-CD14 depletion. To clarify this, a bracket was added between mock vs. PVSRIPO, and the statistical comparison is stated in the figure legend (**revised Figure 1h; revised manuscript, pg. 25**).

16. Fig. 2a: What do blue vs. red asterisks indicate? The legend defines a black asterisk.

We have added this information and apologize for the omission in the original manuscript. Blue and red asterisks apply to statistical significance for PVSRIPO or LPS-related effects, respectively (**revised Figure 2a; revised manuscript, pg. 26**).

17. Material and Methods: What was the sex and age of the mice that were used in the studies? What equipment was used to make tissue slices?

The reviewer is right to point this out; we have now added this important information to the Methods section. Male and female mice were used in the B16 experiments (experiments were sex-matched between groups), and female mice only were used for E0771 and MDA5 -/- vs. wt comparison experiments (revised manuscript, pg. 37).

18. How long did tissue slices remain viable?

We only maintained slice cultures *ex vivo* for up to 48 hrs. Viability (7-AAD) measures in Figure 1c for mock-treated samples showed that 4/6 specimens had >60% viability at 48 hrs, though two specimens were only ~40% viable.

19. There were too many additional minor issues to document in a timely fashion. The manuscript (especially figure and supplementary figure legends) should be revised to add clarity for readers.

We have carefully reviewed and revised the manuscript throughout, in particular the figures and figure legends, and asked a colleague not familiar with the study to review the work and help with optimizing data presentation.

Major Issues:

20. There were a lot of references to functionality of T cells throughout the manuscript. However, the assessments determined complex phenotypes that are often associated with improved functions. Direct functional assessments should be performed (e.g. co-culture cytotoxicity assay, depletion of T cells in efficacy studies). This would also help determine whether the T cells that are being assessed are tumor-specific or virus-specific (or have some other specificity).

The reviewer is right. We have replied to this issue as part of our response to the editorial comments (concern nr. 2). For convenience, we copied our reply below (see also pg. 1-2):

We have added new material in the reconfigured Figure 8 (**new Figure 8d-f**) that directly addresses this question using three distinct approaches; two of which are new to the revision:

1. Figure 8a-c was previously included as a part of Figure 7, where the central importance of this data may not have been obvious. We reconfigured the paper, better emphasizing the prominence of the findings, in the **new Figure 8a-c**.

Data shown in **new Figure 8** are IFN- γ ELISpot analyses performed to demonstrate that tumor antigen-specific splenocytes from mRIPO-treated mice secrete higher per-cell amounts of IFN γ compared to those from LPS-treated mice, despite inducing similar numbers of antitumor T cells (**new Figure 8b**). The data also document higher baseline (left panel) and antigen-specific (right panel) cytokine secretion by splenic T cells after mRIPO treatment (**new Figure 8c**).

Thus, this experiment demonstrates that tumor antigen-specific T cells (an MHC class I-specific epitope was used) secrete more IFN γ after mRIPO treatment.

2. Newly performed experiments reported in **new Figure 8d** show the anti-tumor function of T cells from tumor-draining lymph nodes (TDLN), recovered after mRIPO vs. TLR agonist [Poly(I:C) or LPS] treatment of B16-bearing mice. The assay demonstrates that TDLN T cells from mRIPO-treated mice delayed B16 cell growth *in vitro* more effectively than other treatments (**new Figure 8d**; left-most panel). TDLN T cells from mRIPO-treated B16-bearing mice had no effect on non-homologous CT2A tumor cells (**new Figure 8d**; 2nd left panel). Also, PD-L1 expression on B16 cells was uniquely induced by TDLN T cells from mRIPO-treated B16 tumors and failed to occur on CT2A cells (**new Figure 8d**; 2nd right panels). Lastly, IFN γ /IL-2 secretion by TDLN T cells was uniquely induced in mRIPO-treated B16 tumor-bearing mice 7 days after therapy (**new Supplementary Figure 12c**).

This demonstrates that T cells in tumor-draining lymph nodes restrict tumor cell growth more effectively after mRIPO therapy compared to Poly(I:C) or LPS treatment.

3. Most importantly, we added new experiments with adoptive transfer of splenic T cells recovered from B16-bearing mice treated with mRIPO vs. Poly(I:C) or LPS (**new Figure 8e, f**). Splenic T cells from mRIPO-treated mice produced significantly more effective tumor control in treatment-naïve mice implanted with B16 tumors compared to Poly(I:C) or LPS treatment (**new Figure 8e, f**).

In conjunction with polyfunctional phenotypes (Figure 7e) after intratumor mRIPO vs. TLR agonist treatment, our new data sets demonstrate that mRIPO therapy generates more functional antitumor T cell responses, culminating in tumor antigen-specific cytotoxic T cell lymphocyte responses. Data in new Figure 9 unambiguously link this phenomenon to MDA5-TBK1-IRF3 signaling.

21. The authors highlight that the findings are independent of 'overt' oncolysis. However, in the absence of positive controls, data to argue against oncolysis in the manuscript are weak.

We have replied to this issue as part of our response to the editorial comments (concern nr. 3). For convenience, we copied our reply below (see also pg. 3):

We have added the recommended controls and new data, which clarify the issues related to oncolysis, and the measurements thereof, raised by Reviewer 1. This includes:

1. controls added to measurements of overall cell viability in the tumor slice assay (see **revised Figure 1C**). We added viability data from 3 established GBM cell lines infected with PVSRIPO as a positive 'oncolysis' control. This puts the relative lack of overt cell lysis in the tumor slices in perspective.
2. We added comprehensive analyses of viral translation (**new Figure 3b**), the innate antiviral host response (**new Figure 3b-d**), and cytotoxicity (**new Figure 3a**), for an assessment of the effects of transgenic CD155 expression on B16 and PEC cells after mRIPO infection.

22a. Regardless, why was a model with relatively weak oncolysis used?

22b. What would the results be if the virus were studied in a model in which robust oncolysis occurs? In contrast, what would the results look like if the virus were to be irradiated (rendered replication incompetent). A comparison between the virus and ligands for pattern recognition receptors seems irrelevant if the virus can utilize oncolysis (even if it is limited) to stimulate immune responses by other mechanisms.

We have added **new Figure 3a-d** to more comprehensively introduce our immunocompetent murine tumor models, which we use to define the relative contributions of oncolysis or the non-malignant TME in mediating PVSRIPO therapy. Human CD155 is required for polio (and PVSRIPO) entry in murine cells. Thus, human CD155-tg mouse models that were originally devised for studying polio CNS pathogenicity were used in this study, wherein the *PVR* (CD155) gene is controlled under its own promoter, CD155 expression mirrors that of the human context, and expression of CD155 permits polio entry and recapitulates poliomyelitis disease after WT polio inoculation^{3,4}. We have added data demonstrating that viral translation and oncolysis only occurs in the presence of CD155 expression in murine B16 tumor cells (**new Figure 3a, b**), and that mRIPO leads to cytokine secretion and type-I IFN responses in murine PEC only in CD155-tg mice; UV-inactivated mRIPO does not elicit these responses (**new Figure 3c, d**), indicating that viral replication is required for inflammatory activation of myeloid cells. Using wt (non-permissive to mRIPO) and CD155-tg tumor cells and/or CD155-tg mice, we compared the antitumor efficacy of mRIPO in a model with relatively efficient oncolysis, mediated by transgenic expression of CD155 in neoplastic (B16) cells (see **new Figure 3b**), vs. assays in a model that is categorically resistant to oncolysis (due to absence of human CD155 on B16 neoplastic cells; **Figure 3e**). These experiments, along with confirmation that infection of non-malignant myeloid compartments in tumors mediated immunotherapy (Figure 2 and Supplementary Figure 5), directly implicated infection of the non-malignant TME in PVSRIPO therapy efficacy. This was in line with a lack of observed overt cancerous cell killing (**revised Figure 1c**) in the presence of robust inflammatory cytokine production (Figure 1d) in human

tumor tissue slices. We added explanations in our revised manuscript to better delineate the significance of the findings reported in Figure 3e (**revised manuscript, pg. 7-8**).

Therefore, our rationale for exploring a model system with categorical lack of oncolysis (CD155-tumor cells) stems from our findings that the antitumor efficacy of mRIPO largely depended on infection of the non-malignant TME (Figures 2, 3). Using huCD155-negative tumor models helped to eliminate any confounding effects of oncolysis in our study. In that context, we felt it was very important and appropriate to compare PVSRIPO to standard PRR agonists, used with the intent of eliciting host innate inflammatory responses in the non-malignant TME. This approach was pivotal for defining MDA5-TBK1-IRF3 signaling and its role in Th1-type inflammation and the stimulation of polyfunctional antitumor T cell responses (Figure 9).

However, we also employed mouse models with active oncolysis in our later investigations (B16 melanoma expressing human CD155; **Figure 8a-c**). These showed that, as observed for tumor-infiltrating T cell phenotype (**Figure 7e**), *in vitro* tumor-draining lymph node co-culture assays (**new Figure 8d**), and adoptive transfer studies (**new Figure 8e**), mRIPO induced antitumor T cells with superior IFN γ secretion after B16 cell co-culture as compared to LPS (**Figure 8a-c**).

The method used to culture murine DCs was not described and this is very important with respect to drawing conclusions regarding DC biology. If not sorted, standard "DC cultures" are actually dominated by macrophages and other cells (see: <https://doi.org/10.1016/j.immuni.2015.05.018>). The authors need to prove that they were studying DCs.

The reviewer is correct about the importance of these methodological principles and we have added empirical detail to our Methods section to clarify how BMDCs were derived, eg. for the experiments described in Figure 2d (see **revised manuscript, pg. 37**):

'BMDCs were derived from bone marrow as previously described⁵; briefly, bone marrow cells isolated from femurs and tibias were cultured at a density of 2.5×10^6 cells/ml in RPMI + 10% FBS containing 300ng/ml human recombinant FLT3L (Thermo-Fisher) for 9 days and non-adherent cells were used. FLT3L-derived BMDCs were tested for CD11c expression.'

The methods used for BMDC generation is from Hildner et al (*Science*, 2008⁶); and we apologize for not providing more clarity on the methods related to this in the original submission. **Figure R2** shows flow cytometry data demonstrating CD11c, CD11b, and MHC-class II expression in BMDCs used in Figure 2d:

Figure R2. FLT3L-derived BMDCs were stained for CD11b, CD11c, and IA/IE to confirm IA/IE (MHC-class II) and CD11c expression after differentiation *in vitro* for 9 days in FLT3L-containing media.

Of note, experiments describing DCs in Figures 4-6 were performed using human monocyte-derived DCs, generated by addition of GM-CSF and IL-4 to cultures for 6 days.

Conclusions were drawn regarding infected cells (e.g. macrophages and DCs) but no data were provided to show that these cells were infected. They were treated with the virus but many not have been infected.

Poliovirus' natural target tropism, ie. the range of human host cells targeted in humans exposed to oral poliovirus challenge, prominently includes CD11c+ macrophages/DCs⁷. The only other cell type targeted in the gastrointestinal (GI) tract are an unknown population of CD155+ GI epithelial cells. This tropism is determined by CD155 distribution⁸; CD155 is constitutively expressed in all monocytic lineage cells⁹. The human poliovirus receptor received the Cluster Differentiation 155 designation because of this expression in the myeloid cell compartment.

PVSRIPO's capsid is identical to the type 1 Sabin vaccine capsid, a derivative of the wild type 1 (Mahoney) reference strain. Both wt and Sabin poliovirus strains have identical tropism for- and entry processes mediated by CD155 binding. Therefore, the natural tropism of all polioviruses for macrophages/DCs, mediated by CD155, is shared by PVSRIPO.

Given that it is a natural, high priority target of poliovirus in live primates with powerful barriers to infection (eg. a functioning immune system, the GI mucosa, etc.)⁷, it is expected that treatment of tumor slices or intratumor inoculation of tumors implanted in CD155-tg mice, also target the macrophage/DC compartment.

Directly demonstrating PVSRIPO infection of macrophages/DCs is difficult, because PVSRIPO has a non-cytopathogenic, lingering propagation phenotype in such cells^{5, 10}. This phenotype is associated with low-level, transient viral translation and low levels of ongoing viral replication that make detection of infected monocytic lineage cells, eg. by immunohistochemistry for viral proteins difficult. In contrast, wt poliovirus infection of macrophages/DCs in susceptible primates produces rampant, cytopathogenic viral protein synthesis, which can be readily detected by IHC⁷. The primary outcome of PVSRIPO infection of macrophages/DCs is a powerful, sustained host innate antiviral immune response⁵. Therefore, we have obtained compelling indirect evidence for PVSRIPO targeting of tumor-associated macrophages by documenting IFIT1 induction in such cells by flow cytometry (Figure 1g).

To obtain direct evidence for viral targeting of tumor-associated macrophages, we attempted detection of viral dsRNA with the 'K1' antibody (Scicons; recognizes dsRNA intermediate of viruses). K1 antibody staining was included in the staining panel for data shown in Figure 1g

(this is noted in the methods section). We were not able to detect specific K1 staining beyond that of the isotype control. This is almost certainly due to the very low levels of viral propagation in macrophages/DCs infected with PVSRIPO, producing insufficient amounts of viral dsRNA replication intermediates for detection by K1 antibody. The virus dose used in these assays (1×10^8 pfu) matches the clinical dose administered in GBM and melanoma patients.

In the context of these assays, we cannot ensure that every macrophage/DC present in the tissue slice was infected. The goal of the slice culture assays was not to decipher the percentage of cells productively infected with PVSRIPO, but rather to determine the bulk effect of PVSRIPO treatment in a complex tissue recapitulating the heterogeneous composition of GBM, melanoma and breast cancer tissue not skewed by dissociation and serial passage culture. We have provided compelling evidence for glioma-associated macrophages to be the primary compartment responding to PVSRIPO treatment. This is in perfect alignment with the role of macrophages/DCs as key poliovirus targets in natural infections after oral exposure⁷. We deciphered the innate immune stimulatory mechanisms induced by PVSRIPO infection at great depth, using primary murine and human myeloid cell cultures.

For mouse peritoneal exudate cell (PEC) experiments (Figures 2, 3): we were unable to detect mRIPPO translation by immunoblot in PECs (**new Figure 3d**). This is in alignment with earlier analyses, where we documented the transient, low-level translation phenotype of PVSRIPO in primary human monocyte-derived macrophages¹⁰. We have now added **new Figure 3c** which documents type-I IFN responses only in PECs expressing CD155 after infection with PVSRIPO. In this assay, infection of CD155- PEC or infection of CD155+ PEC with UV-inactivated PVSRIPO failed to induce innate antiviral IFNs (**new Figure 3c**). Also, only PEC infected with PVSRIPO mounted a phospho-STAT1(Y701) signaling response (**new Figure 3d**). From these data we conclude that productive infection, ie. CD155-mediated PVSRIPO entry and production of viral RNA intermediates inducing host innate antiviral immunity, is required for PEC activation. A multiplicity of infection (MOI) of 10 was used in these experiments to ensure homogeneous % infection of all cells in the culture. This is the accepted standard in the field, based on overwhelming evidence from >60 years of experimentation with poliovirus in tissue culture.

For human monocyte-derived macrophages/DCs (Figures 4-6): we detected viral protein (2C), which is a non-structural viral protein only produced during active viral replication (Figure 5a; Figure 6a; Supplemental Figure 9d). We also confirmed that, as in mouse PEC (**new Figure 3d**), human monocyte-derived macrophage activation/cytokine secretion does not occur after treatment of cells with UV-inactivated PVSRIPO (Supplementary Figure 6g). This indicates that viral replication, producing viral dsRNA replication intermediates, is required to induce host innate immunity, as has been established previously⁴. An MOI of 10 was used in all *in vitro* studies of human monocyte-derived macrophage/DCs to ensure homogeneous infection of all cells in the culture (see above).

For *in vivo* study (Figures 3, 6-9): we confirmed that CD155 expression was categorically required for the antitumor efficacy of PVSRIPO in mice, indicating that active viral infection is required for the observed effects.

In summary, while—for technical reasons—it is exceedingly difficult to ascertain percentages of cells harboring productive PVSRIPO translation/replication, we have provided overwhelming

empirical evidence that active viral infection/propagation, mediated by CD155, is categorically required for PVSRIPO's antitumor effects in all assay systems.

A mechanism of action for oncolytic poliovirus was proposed. How do the authors envision harnessing such knowledge to improve virotherapies for cancers?

We have added a conclusion statement in the final paragraph of the Discussion to explain the relevance of our discoveries to clinical virotherapy. The statement is pasted below for convenience:

'Altogether, our findings indicate a distinct mechanism by which polio virotherapy reinvigorates immune-surveillance and mediates antitumor efficacy. This mechanism depends upon TBK1-IRF3 signaling in the non-malignant tumor microenvironment. Thus, future investigation into how TBK1-IRF3 signaling in the TME might be accentuated, sustained, or complemented is warranted to determine how to improve the efficacy of virotherapy' (**revised manuscript, pg. 19**).

Reviewer #2 (oncolytic viral vectors) (Remarks to the Author):

What are the major claims of the paper?

- *That recombinant poliovirus RIPO therapy primarily mediates antitumor immunotherapy via direct infection of the non-malignant tumor microenvironment, independent of overt malignant cell lysis.*
- *Relative to other innate immune agonists, virotherapy provoked selective, exaggerated TBK1-IRF3 driven innate inflammation that was associated with sustained type-I/III interferon (IFN) release.*
- *Despite priming equivalent antitumor T cell quantities, virus-mediated TBK1-IRF3 signaling, but not NFkB-polarized TLR activation, culminated in polyfunctional and Th1-differentiated antitumor T cell phenotypes.*
- *Recombinant type-I IFN bolstered tumor-localized T cell function but did not mediate durable antitumor immunotherapy without concomitant pattern recognition receptor (PRR) signaling. Thus, virus-induced TBK1-IRF3 signaling in the TME provides PRR-contextualized IFN responses that elicit functional antitumor T cell immunity.*
- *TBK1-IRF3 innate signal transduction determines eventual function and differentiation of tumor infiltrating T cells.*

Are the claims novel? If not, please identify the major papers that compromise novelty.

1. The novelty of the manuscript is limited . For example, “STING senses the presence of nucleic acids from intracellular pathogen infection and in turn initiates a downstream signaling cascade that includes TBK1-mediated activation of IRF3 and Stat6, resulting in IFN-I and cytokine production.” J Cell Commun Signal. 2018 Mar; 12(1): 83–90.

Evaluating TBK1 as a Therapeutic Target in Cancers with Activated IRF3 DOI: 10.1158/1541-7786.MCR-13-0642 Published July 2014.

It is true that the role of type-I IFNs in T cell priming is well established. We believe our work provides a fundamental, important advance in the understanding of how innate antiviral immunity may be leveraged for cancer immunotherapy, eg. through recombinant RNA viruses:

1. The primary target and mediator of antitumor efficacy of PVSRIPO therapy is the non-malignant TME (Figures 1-3). This is a key distinction, as the field of ‘oncolytic viruses’ has predominantly focused on targeting of- and virus:host interactions playing out in neoplastic cells.
2. Despite a status as canonical type-I IFN inducers in tumors, Poly(I:C) (TLR3 agonist), LPS (TLR4 agonist), and 2’3’-cGAMP (STING agonist)—derivatives of which are being pursued as cancer immunotherapies—do not effectively induce type-I IFN responses in primary human tissue and human myeloid cells when compared to PVSRIPO (Figures 1 and 4). Our work shows that engaging distinct PRRs, ie. the cytoplasmic dsRNA sensor MDA5 vs. cell surface TLR agonists, elicits differentiated innate inflammatory signatures that result from divergent engagement of the innate kinase assemblies anchored on TBK1/IKK ϵ or on IKK α/β . We provide compelling evidence that intracellular dsRNA, either in the form of viral PVSRIPO RNA or Poly(I:C) complexed with PEI/lipofectamine (Figures 1 and 4), elicits far stronger type-I IFN induction in primary human tumor tissues, and mediates substantially better antitumor efficacy in murine models (Figure 9g).

3. We demonstrated through a comprehensive series of *in vivo* studies in mouse tumor models that the distinctive innate inflammatory footprint resulting from selective MDA5-IRF3-TBK1-IFN signaling ultimately dictates the functional capacity of tumor infiltrating T cells, in particular their capacity for T-bet and GzmB expression. Thus, the specific MDA5-IRF3-TBK1-IFN signaling emanating from PVSRIPO infection of the non-malignant TME instigates tumor antigen-specific antitumor immunity (Figures 7-9).
4. Our work elucidates the pivotal importance of the inflammatory context of type-I IFNs for efficacious stimulation of T cell *function*. While the critical roles of type-I IFN in the priming and function of T cells are well documented, our study elucidates the essential contributions provided by the broad proinflammatory stimulus of PRR signaling/innate inflammation that contextualizes type-I IFN. This distinction is demonstrated in Figure 9c-e, where we show that 1) removing type-I IFN signaling impedes the antitumor efficacy of Poly(I:C) (which induced intratumor IFN in mice; Figure 7c) and supplementing LPS (which did not induce IFN in mice; Figure 7c) with recombinant type-I IFN led to substantial antitumor efficacy.
5. Respectfully, we would also like to note that both papers referenced by the reviewer are on subject matter at best tangential to our core conclusions. Ref. #1 merely reports that activating STING engages TBK1-IRF3. Ref. #2 proposes to *inhibit* TBK1 for cancer therapy in circumstances where IRF3 signaling is constitutive.

Thus, our observations fundamentally advance insight into effectively engaging the notoriously immunosuppressive tumor microenvironment for cancer immunotherapy, and the mechanistic understanding of cancer-targeting recombinant viruses in particular.

2. *Will the paper be of interest to others in the field?*

Yes.

3. *Will the paper influence thinking in the field?*

Not markedly. Aside from modest novelty, another reason is that effects seen in Figure 2 are modest. Infection with mRIPO of PEC, BDMC or non-malignant TME has modest, though significant, anti-tumor effects. Likewise, for Figure 6A. Figure 7C shows that the most efficacious effect is seen by Poly(I:C)-PEI. In 7E LPS + IFN α is most efficacious.

Please see our statement to point #1. We agree with the reviewer that the antitumor efficacy of mRIPO in mice is modest generally. As mentioned in our Discussion, this is due to host range issues (polio is an exclusive human pathogen; old-world primates can be infected experimentally). For example, analysis of tumor slice cultures from ex vivo human tumors (Figure 1) revealed substantial type-I IFN and CXCL10 secretion; yet, in murine B16 tumor homogenate no significant cytokine induction was observed, with CXCL10 only moderately induced beyond background (Figure 7c).

We believe that the rigor of the models used also is a factor in this phenomenon. B16 (a murine melanoma model) is refractory to PD1 blockade in our investigations; yet, anti-PD1 became

standard-of-care therapy for melanoma. B16 is notoriously resistant to cancer immunotherapy generally, and grows remarkably fast.

Beyond experiments performed with mRIPO/PVSRIPO, we provide compelling proof-of-principle evidence that MDA5-TBK1-IRF3 signaling and the type-I IFN dominant inflammation this induces is *capable* of mediating robust antitumor efficacy. These efforts are shown in the combination of LPS+IFN α as well as Poly(I:C)-PEI (Figure 9g); the latter of which was shown to be dependent upon MDA5 signaling (Figure 9h, i). Lastly, while we acknowledge that the impact of this work is in part due to the potential clinical utility of PVSRIPO, our aim was to demonstrate the key biological role of MDA5-TBK1-IRF3 signaling in the TME in selectively determining the antitumor *function* of T cells.

Are the claims convincing? If not, what further evidence is needed?

The data provided is of high-quality and credible. The major problems remain novelty and efficacy. Are there other experiments that would strengthen the paper further? How much would they improve it, and how difficult are they likely to be?

4. Additional experimentation might provide an experimental protocol that might be more efficacious and enhance the potential of the suggested approach using viral infection of the tumor microenvironment via selective Tbk1-Irf3 signaling. These are no likely to be easily accomplished in a short period of time.

In essence, we have begun to assess what the reviewer is suggesting in experiments reported in the **new Figure 9** and in planned new investigations that are beyond the scope of the present manuscript (see our response to Rev. #3; pg. 18). The primary empirical goals of this manuscript are mechanistic: to elucidate the quality and character of innate signaling in the TME that ultimately produces effective antitumor T cell immunity. We agree that mRIPO/PVSRIPO does not accomplish this task robustly in our mouse tumor models; the intensity of TBK1-IRF3 signaling in B16 tumors is not high (Figure 8c). Yet, the *pattern* of innate activation in mouse myeloid cells is consistent between mouse and human systems, rendering mouse models valid tools for our mechanistic work. This is also evident through our work on synthetic PRRs, which are not affected by host species-specific innate immune factors that affect PVSRIPO research in mouse models. Cytoplasmic dsRNA (which mimics viral dsRNA intermediates), eg. Poly(I:C)-PEI, induces MDA5-TBK1-IRF3 signaling in the TME and mediates robust antitumor efficacy/antitumor T cell responses as shown in **new Figure 9f-i**. These proof-of-principle assays, which are not designed to maximize therapy effect in contrived mouse tumor models, achieves our goal of advancing the mechanistic understanding of effectively engaging the tumor microenvironment for immunotherapy.

Are the claims appropriately discussed in the context of previous literature?

There is discussion of the literature, though the novelty of the approach give studies by other groups might be overstated.

We have revised our manuscript to properly highlight the new findings of our work, and to explain their significance in light of previously published evidence. For example, we included discussion of prior work demonstrating that type-I IFN directly potentiates T cell function, and documenting the role of MDA5 in mediating effective antiviral T cell responses. We are not aware of: 1) prior cancer virotherapy research demonstrating a TME-dependent mechanism that

mediates antitumor efficacy ; 2) previous virotherapy research employing primary *ex vivo* tumor slices to compare viral vs. PRR agonist inflammatory signatures; 3) evidence of the unique sustained MDA5-IRF3-TBK1- signaling phenotype and myeloid cell activation patterns induced by cytoplasmic dsRNA vs other PRR agonists; nor 4) empirical demonstration that proper antiviral PRR signaling + recombinant type-I IFN provides an optimal context for priming of antitumor T cell immunity.

If the manuscript is unacceptable in its present form, does the study seem sufficiently promising that the authors should be encouraged to consider a resubmission in the future?

The data are sound, and the manuscript is well written. The manuscript appears to have a modest impact. Not certain that this impact can be markedly enhanced in the near term.

Is the manuscript clearly written? If not, how could it be made more accessible?

Could the manuscript be shortened to aid communication of the most important findings?

Have the authors done themselves justice without overselling their claims?

The manuscript is well written, but impact is overstated.

We respectfully disagree with the reviewer that our work lacks novelty; please see our responses on pg. 13-15 above. We have strived to emphasize the innovative aspects of our work in the revised manuscript.

Have they been fair in their treatment of previous literature?

Greater discussion of related findings by others would provide a more comprehensive background for the reader.

We have revised our Discussion to provide a more comprehensive interpretation of our data in the light of previously published findings (**revised manuscript, pg. 17-19**).

Have they provided sufficient methodological detail that the experiments could be reproduced?

Yes.

Is the statistical analysis of the data sound?

Yes.

Should the authors be asked to provide further data or methodological information to help others replicate their work? (Such data might include source code for modelling studies, detailed protocols or mathematical derivations).

Not necessary.

Are there any special ethical concerns arising from the use of animals or human subjects?

There are no apparent concerns.

Reviewer #3 (tumour microenvironment) (Remarks to the Author):

The ms by Brown et al investigates virotherapy by a recombinant poliovirus (PVSRIPO) that shows promise in clinical trials as a means to combat solid tumors via stimulating tumor immunity. This study reports that improved antitumor immunity using this method is driven by innate immune responses by non-malignant TME cells (including a strong response by TAMs) and not lysis of tumor cells themselves. This was seen for GBM, breast and melanoma indicating a tumor type independent response. Further, this response is characterized by an MDA5-TBK1-IRF3 mediated IFN response, that when pharmacologically inhibited, reduced the efficacy of this virotherapy in mice. Of interest, IFNs or PAMPs alone were not sufficient to drive this response indicating a distinct signal sent by the live virus during its replication. Further, recombinant poliovirus therapy triggered a robust T cell response that was characterized by the expansion of polyfunctional effector T cells.

This is an interesting study and a well written paper that makes use of primary tumor sections and a newly established mouse model of PVSRIPO infection (mRIPO, CD155 Tg mice) to interrogate the mechanistic basis for this specific form of virotherapy against solid tumors. The experiments are well done and data clearly presented, and the work collectively provides important insights into how PVSRIPO is able to bolster antitumor immunity.

Despite my enthusiasm, some minor weaknesses include the following:

1. Models whereby MDA5, TBK1 and/or IRF3 are genetically lacking from TAMs and other cell types within the TME would provide stronger mechanistic support for the conclusions.

We thank the reviewer for this suggestion, and have followed her/his recommendation. We now confirm our findings using MDA5 knockout mice in **new Figure 9h** and **i**, in assays using Poly(I:C)-PEI in a rodent tumor model lacking MDA5 in the non-malignant TME.

Poly(I:C)-PEI recapitulates the signature innate inflammation induced by PVSRIPO (see Figure 4c-g and Figure 5c, d). In MDA5^{-/-} mice implanted with wt B16 tumors, only the non-malignant TME is devoid of MDA5. Immunotherapy efficacy of Poly(I:C)-PEI was blocked in MDA5^{-/-} mice (**new Figure 9h**) and the induction of T-bet and GzmB-expressing CD4 and CD8 T cells was diminished after therapy (**new Figure 9i**).

We did not carry out investigations with PVSRIPO in MDA5^{-/-} models, for several reasons. This would have entailed generating a homozygous MDA5^{-/-} mouse colony transgenic for human CD155. The crossbreeding/build-up required to create such a colony would entail very long delays, which are exacerbated by COVID19-related restrictions in our Lab Animal facilities. Fortunately, we had begun breeding MDA5^{-/-} mice prior to COVID19 in anticipation of future studies in this model. Also, enteroviruses related to poliovirus (e.g. Coxsackie B virus), unsurprisingly, were reported to have pathogenicity phenotypes in MDA5^{-/-} mice¹¹. Thus, Poly(I:C)-PEI was appropriate for our proof-of-principle investigations, as it recapitulated the MDA5-TBK1-IRF3 dominant inflammatory phenotypes (Figures 4 and 5), similarly induced robust polyfunctional antitumor T cell phenotypes associated with antitumor efficacy (**Figure 9g**, **Supplementary Figure 14b**), does not cause toxicity in the absence of MDA5 as would be anticipated by enterovirus infection, and could be tested in the native MDA5^{-/-} mice already at hand.

In addition, we have extended our Amlexanox (TBK1 inhibitor; AMX) data set to demonstrate that the antitumor effect of the PVSRIPO-inflamed TME is dependent on TBK1 signaling (**new Figure 9**). Combining PVSRIPO therapy with AMX eliminated the treatment effect in the B16 murine tumor model (**new Figure 9b**) and hindered the induction of GzmB/T-bet expressing CD4 and CD8 T cells (**new Figure 9a**).

Therefore, we believe that our investigations have provided compelling mechanistic evidence that MDA5-IRF3-TBK1-type-I/III IFN dominant inflammation is the core element of PVSRIPO's antitumor effect. Our complementary data sets on the role of MDA5-TBK1-IRF3 signaling in both innate activation (Figures 5, 6), and ensuing adaptive stimulation (Figure 9) after PVSRIPO treatment/therapy, provide compelling documentation of MDA5-TBK1-IRF3 signaling-induced antitumor T cell immunity (Figure 9).

2. However, the TBK1 inhibitor and IFNAR blocking Ab experiments do provide some functional data that support their proposed model of PVSRIPO antitumor function, and should suffice. While the TBK1 inhibitor (AMX) did reduce GzmB CD8+ T cells, tumor growth data from this experiment should also be shown, as it would be expected that tumor growth would increase.

The reviewer is right. We added tumor volumes/clinical monitoring data at the time of tissue harvest from the AMX/IKK16-inhibitor experiments reported in the original submission (**new Figure 9b**). The repeat, confirmatory experiment combining AMX with mRIPO on tumor growth is shown in **Figure R3**. As the reviewer predicted, AMX treatment prevented antitumor effects of mRIPO treatment (**new Figure 9b; Figure R3**):

Figure R3. Repeat assay combining mock or mRIPO +/- AMX inhibitor as in main text **new Figure 9b**.

3. It would strengthen the argument that heightened and sustained TBK1-IRF3 is the critical driver of the antitumor response mediated by PVSRIPO if the authors could devise a gain of function approach to specifically increase and sustain this pathway in the TME.

We have launched an investigation in this direction that we believe will extend beyond the scope of this report, due to the extensive validation and mechanistic work that is required. We appreciate the reviewer propelling this new investigation forward, which we intend to build upon to identify novel combination drug candidates to improve virotherapy. We have discovered an approach that synergistically enhances type-I/III IFN responses in monocyte-derived macrophages after PVSRIPO infection or after Poly(I:C) transfection in multiple donors. This is based on targeting a recently described mechanism of IRF3 inhibition, mediated by a negative feedback loop. However, due to limited available human CD155-tg mice for testing *in vivo*—our colony production has only recently rebounded from COVID19-related shutdowns—and technical issues related to drug solubility for *in vivo* administration have restricted our efforts to move this project forward.

Since the gain of function effect for our study must be inducible, and acting selectively in the TME (and not in the myeloid compartment at large), it would be extremely difficult to devise.

We believe that our conclusion that MDA5-TBK1-IRF3 signaling (MDA5, uniquely, is the dominant PRR recognizing picornaviruses like PVSRIPO¹²; picornaviruses are not sensed by RIG-I) selectively induces a functional antitumor T cell response is now solidly supported through multiple complementary approaches that encompass both PVSRIPO and model PRR agonist treatments:

1. TBK1-IRF3 signaling is selectively induced by PVSRIPO or by cytoplasmic dsRNA, ie. in the form of transfected Poly(I:C) (**Figures 4, 5**)
2. Human macrophage activation by PVSRIPO occurs via MDA5-TBK1-IRF3 signaling (**Figure 6**)
3. PVSRIPO infection of murine cells/tumors results in a TBK1-IRF3 dominant phenotype (**Figure 2a**), similar to that of human myeloid cells/tumors, that is inhibited by AMX, the TBK1 inhibitor (**new Supplementary Figure 13a**).
4. mRIPO infection of the non-malignant TME, which is defined by a TBK1-IRF3 dominant innate host response, culminates in antitumor T cell phenotypes with greater function than that induced by PRR agonists Poly(I:C) or LPS (**Figure 7; new Figure 8d-f**).
5. The efficacy of mRIPO therapy in potentiation of T cell functional phenotypes is dependent upon TBK1 signaling (**Figure 9a; new Figure 9b**).
6. Type-I IFN, in the context of PRR signaling, mediates antitumor efficacy and T cell activation in tumors (**Figure 9d-e**).
7. Cytoplasmic dsRNA [Poly(I:C)-PEI] induces type-I IFN-dominant macrophage activation in mouse tumor models similar to human cells (**Figures 4, 5**), and mediates superior antitumor efficacy compared to 'plain' (extracellular) Poly(I:C) in an MDA5-dependent manner (**new Figure 9g-i**).

4. Can the authors elaborate further on how they think PVSRIPO is able to drive polyfunctional T cell phenotypes, and why the PAMPs did not?

We have revised our manuscript in the Discussion to better explain our central argument why we think that MDA5-TBK1-IRF3-dominant inflammation instigated by PVSRIPO/cytoplasmic dsRNA induces polyfunctional T cell phenotypes (**revised manuscript, pg. 18-19**). We have copied the main revised section of the Discussion below:

'PVSRIPO proficiently induces sustained TBK1-IRF3, type-I/III IFN-driven inflammation in human myeloid cells and primary human tumor tissues. Yet, mRIPO therapy evoked only modest pro-inflammatory cytokine induction in murine B16 tumors relative to LPS/Poly(I:C). Nonetheless, T cell polyfunctionality was higher after mRIPO treatment, in a TBK1-IRF3 dependent manner. Also, mimicking viral dsRNA by cytoplasmic delivery of Poly(I:C), induced prominent TBK1-IRF3 signaling and mediated superior antitumor immunotherapy relative to uncomplexed Poly(I:C) in an MDA5-dependent manner. Our findings extend the role of sustained, MDA5-dependent, type-I IFN in determining functional CD8 T cell responses to LCMV

infection¹³; they indicate the therapeutic potential of MDA5 signaling/RNA virus replication in the TME to efficiently stoke functional antitumor T cells. Since viral/dsRNA-induced MDA5 signaling led to pronounced, sustained IFN responses in the context of a broader inflammatory program (e.g. myeloid cell activation and immune cell influx), and IFN was critical for the functional status of TILs, we conclude that virotherapy elicits functional antitumor T cell immunity via PRR signals that encompass IFN. Our study identifies a cell-signaling nexus where PRR-specific signals (e.g. TLR4 vs. MDA5) impact distinct innate phenotypes and eventual T cell fate: via tunable regulation of TBK1-IRF3 vs. IKK α/β -NF κ B innate kinase assemblies.'

We believe that Figure 9 most clearly demonstrates why PVSRIPO induces stronger polyfunctional T cell phenotypes in tumors (Figure 7e) and induces antitumor T cells with higher IFN γ secreting capacity (Figure 8c). In Figure 9, supplementing LPS with recombinant IFN provides substantial antitumor efficacy and enhances TIL activation phenotypes (Figure 9d-e). Blocking type-I IFN signaling with IFNAR antibody in combination with Poly(I:C) therapy mitigates its antitumor efficacy (Figure 9c). *Cytoplasmic* Poly(I:C)-PEI, which partially recapitulates the type-I/III IFN signature induced by PVSRIPO (Figures 4-5; Figure 9f), is superior to plain Poly(I:C) in mediating antitumor efficacy and inflaming tumors with T cells (**new Figure 9f; Supplementary Figure 14b**). Cytoplasmic Poly(I:C), eg. after transfection in cells, is known to exert MDA5 activation, while plain Poly(I:C) mainly engages cell surface TLR3, has been shown to induce sustained IFN release during viral infection¹³, explaining the innate phenotype that defines the myeloid cell response to PVSRIPO (Figure 6a, b). Without MDA5, cytoplasmic Poly (I:C) does not mediate substantial antitumor efficacy or cause enhanced functional (T-bet and GrzmB) tumor infiltrating T cell phenotypes. Collectively, these data demonstrate the critical importance of PRR contextualized, sustained type I IFN response and highlight a role for MDA5-induced IFN in engaging functional T cell responses. Thus, in short, we conclude that the presence of sustained type I IFN within the context of PRR signaling is an optimal context to induce functional T cell responses.

5. Statistical significance is not demonstrated for some figures, and this makes it difficult to determine if some differences, such as differences in tumor growth in figures 2, 6 and 7, are meaningful. This should be added. Further, number of experimental replicates should be noted in the legend for each figure.

The reviewer is right. This information has been added to figures and figure legends where it was not provided before.

REFERENCES

1. Alexopoulou, L., Holt, A.C., Medzhitov, R. & Flavell, R.A. Recognition of double-stranded RNA and activation of NF-kappaB by Toll-like receptor 3. *Nature* **413**, 732-738 (2001).
2. Gromeier, M., Lachmann, S., Rosenfeld, M.R., Gutin, P.H. & Wimmer, E. Intergeneric poliovirus recombinants for the treatment of malignant glioma. *Proc Natl Acad Sci U S A* **97**, 6803-6808 (2000).
3. Desjardins, A. *et al.* Recurrent Glioblastoma Treated with Recombinant Poliovirus. *N Engl J Med* **379**, 150-161 (2018).
4. Feng, Q. *et al.* MDA5 detects the double-stranded RNA replicative form in picornavirus-infected cells. *Cell Rep* **2**, 1187-1196 (2012).

5. Mosaheb, M.M. *et al.* Genetically stable poliovirus vectors activate dendritic cells and prime antitumor CD8 T cell immunity. *Nat Commun* **11**, 524 (2020).
6. Hildner, K. *et al.* Batf3 deficiency reveals a critical role for CD8alpha+ dendritic cells in cytotoxic T cell immunity. *Science* **322**, 1097-1100 (2008).
7. Shen, L. *et al.* Pathogenic Events in a Nonhuman Primate Model of Oral Poliovirus Infection Leading to Paralytic Poliomyelitis. *J Virol* **91** (2017).
8. Iwasaki, A. *et al.* Immunofluorescence analysis of poliovirus receptor expression in Peyer's patches of humans, primates, and CD155 transgenic mice: implications for poliovirus infection. *J Infect Dis* **186**, 585-592 (2002).
9. Freistadt, M.S., Fleit, H.B. & Wimmer, E. Poliovirus receptor on human blood cells: a possible extraneural site of poliovirus replication. *Virology* **195**, 798-803 (1993).
10. Brown, M.C. *et al.* Cancer immunotherapy with recombinant poliovirus induces IFN-dominant activation of dendritic cells and tumor antigen-specific CTLs. *Sci Transl Med* **9** (2017).
11. Wang, J.P. *et al.* MDA5 and MAVS mediate type I interferon responses to coxsackie B virus. *J Virol* **84**, 254-260 (2010).
12. Kato, H. *et al.* Differential roles of MDA5 and RIG-I helicases in the recognition of RNA viruses. *Nature* **441**, 101-105 (2006).
13. Wang, Y. *et al.* Timing and magnitude of type I interferon responses by distinct sensors impact CD8 T cell exhaustion and chronic viral infection. *Cell Host Microbe* **11**, 631-642 (2012).

REVIEWERS' COMMENTS

Reviewer #1 (Remarks to the Author):

The authors responded thoroughly to the comments provided by reviewers. As a result, the manuscript has been strengthened substantially. In this reviewer's opinion, the manuscript is acceptable for publication.

Reviewer #2 (Remarks to the Author):

The manuscript entitled "Viral Infection of the Tumor Microenvironment Mediates Antitumor Immunotherapy via Selective TBK1-IRF3 Signaling" is well written and the data included convincing and derived from well performed studies.

The field of virotherapy has extensively described the role of innate immune activation as well as dendritic cells and interferon responses. It is unclear to this reviewer that the observations reported are sufficiently novel to contribute in a more than an incremental manner to our existing knowledge. This manuscript might be best suited for a specialty journal focused on virotherapy.

Reviewer #3 (Remarks to the Author):

My concerns have been adequately addressed.

Dear Colleagues,

We would like to thank the reviewers for their constructive reviews and suggestions.

Reviewer #1 (Remarks to the Author):

The authors responded thoroughly to the comments provided by reviewers. As a result, the manuscript has been strengthened substantially. In this reviewer's opinion, the manuscript is acceptable for publication.

We thank the reviewer for her/his expert critique and advice towards improving our work.

Reviewer #2 (Remarks to the Author):

The manuscript entitled "Viral Infection of the Tumor Microenvironment Mediates Antitumor Immunotherapy via Selective TBK1-IRF3 Signaling" is well written and the data included convincing and derived from well performed studies.

The field of virotherapy has extensively described the role of innate immune activation as well as dendritic cells and interferon responses. It is unclear to this reviewer that the observations reported are sufficiently novel to contribute in a more than an incremental manner to our existing knowledge. This manuscript might be best suited for a specialty journal focused on virotherapy.

We agree that innate immunity has been a recent focus of the virotherapy field, and the role of innate immunity, including type I IFNs, in priming antitumor T cells during natural infectious processes as well as PRR agonist therapy is generally well-defined. However:

1. There is considerable disagreement in the virotherapy community as to whether innate immunity should be subverted to enhance oncolysis (e.g. via JAK-STAT inhibitors), or whether it should be accentuated to enhance T cell priming. This is critical to resolve, as it impacts clinical implementation. *Our work clarifies that, at least for a clinically relevant recombinant poliovirus (PVSRIPO), activation of innate immunity (in the TME) mediates overall antitumor efficacy.*
2. The nature of viral-mediated innate immune stimulation vs that of clinically relevant PRR agonists, to our knowledge, has not been compared side-by-side in human tumor specimens, mice, or other model systems. It remained to be shown if the inflammation/ antitumor immunity they engender differ in character, quality, or intensity to any extent. As several PRR agonists (STING agonists, Poly IC-LC, TLR4 agonists, TLR9 agonists, etc) move forward to clinical use, this question is critical if oncolytic viruses are viewed as PRR engagers. *We define a unique signature mediated by sustained TBK1-IRF3 activation after viral-induced MDA5 signaling. This pattern was distinct and more potent in inducing functional antitumor immunity compared to TLR3/4 agonists; indeed, TBK1-IRF3 signaling determined tumor infiltrating T cell function. While alternative routes may mimic the inflammatory effect of virotherapy (e.g. cytoplasmic delivery of Poly IC), these data establish the nature of MDA5, and possibly other cytoplasmic PRRs (e.g. RIG-I and STING), to engage sustained TBK1-IRF3 in the TME that culminates in stronger antitumor T cell responses.*

3. Additionally, while most oncolytic viruses are currently thought of as immune engagers, the extent to which inflammatory cell death/tumor cell infection vs infection of the TME, prior to our study, remained unclear. This has been resolved for STING agonists, where similar to our study, it has been shown that STING signaling in the TME mediates the antitumor efficacy of STING agonists (Sivick K et al *Cell Reports* 2018; PMID: 30540940). *We are not aware of any studies that demonstrate a prominent role of the TME in mediating T cell priming and antitumor efficacy of an oncolytic virus capable of occurring completely independent of oncolysis/tumor cell infection, as shown in our work.*
4. It is incorrect that the question of the role of dendritic cells (DC) has been addressed appropriately in the context of virotherapy before. To the contrary: almost all agents contemplated for cancer immunotherapy are based on viruses with well-known, elaborate strategies to suppress, subvert, or skew DC function, in some instances involving outright DC killing. *Our work helped to decipher a peculiar virus:host relationship in DCs for PVSRIPO by defining its highly specific innate inflammatory imprint in this compartment.*
5. Lastly, the question of how innate immune signaling sculpts downstream adaptive immunity is an active and fruitful area of investigation (for example, see Jain and Pasare *Journal of Immunology*. 2017). While our work focuses on translational aspects of innate signaling after virotherapy/PRR agonists, it also contributes to the understanding of how different PRRs influence distinct adaptive polarizations/activities. Routes known to enable PRRs have been largely focused on the cell-type specific expression of PRRs. *Our work shows how different PRRs drive explicitly distinct TBK1-IRF3 vs. IKK-NF κ B signaling to toggle specific innate activation signatures (Figs 1-6), as well as eventual T cell function (Figs 7-9).*

Reviewer #3 (Remarks to the Author):

My concerns have been adequately addressed.

We thank the reviewer for her/his expert critique and advice towards improving our work.